# Model-Based Domain Generalization

**Alexander Robey**     **George J. Pappas**     **Hamed Hassani**
Department of Electrical and Systems Engineering
University of Pennsylvania
{arobey1,pappasg,hassani}@seas.upenn.edu

## Abstract

Despite remarkable success in a variety of applications, it is well-known that deep learning can fail catastrophically when presented with out-of-distribution data. Toward addressing this challenge, we consider the *domain generalization* problem, wherein predictors are trained using data drawn from a family of related training domains and then evaluated on a distinct and unseen test domain. We show that under a natural model of data generation and a concomitant invariance condition, the domain generalization problem is equivalent to an infinite-dimensional constrained statistical learning problem; this problem forms the basis of our approach, which we call Model-Based Domain Generalization. Due to the inherent challenges in solving constrained optimization problems in deep learning, we exploit nonconvex duality theory to develop unconstrained relaxations of this statistical problem with tight bounds on the duality gap. Based on this theoretical motivation, we propose a novel domain generalization algorithm with convergence guarantees. In our experiments, we report improvements of up to 30% over state-of-the-art domain generalization baselines on several benchmarks including ColoredMNIST, Camelyon17-WILDS, FMoW-WILDS, and PACS. Our code is publicly available at the following link: https://github.com/arobey1/mbdg.

## 1   Introduction

Despite well-documented success in numerous applications [1–4], the complex prediction rules learned by modern machine learning methods can fail catastrophically when presented with out-of-distribution (OOD) data [5–9]. Indeed, rapidly growing bodies of work conclusively show that state-of-the-art methods are vulnerable to distributional shifts arising from spurious correlations [10–12], adversarial attacks [13–17], sub-populations [18–21], and naturally-occurring variation [22–24]. This failure mode is particularly pernicious in *safety-critical applications*, wherein the shifts that arise in fields such as medical imaging [25–28], autonomous driving [29–31], and robotics [32–34] are known to lead to unsafe behavior. And while some progress has been made toward addressing these vulnerabilities, the inability of modern machine learning methods to generalize to OOD data is one of the most significant barriers to deployment in safety-critical applications [35, 36].

In the last decade, the *domain generalization* community has emerged in an effort to improve the OOD performance of machine learning methods [37–40]. In this field, predictors are trained on data drawn from a family of related training domains and then evaluated on a distinct and unseen test domain. Although a variety of approaches have been proposed for this setting [41, 42], it was recently shown that that no existing domain generalization algorithm can significantly outperform empirical risk minimization (ERM) [43] over the training domains when ERM is properly tuned and equipped with state-of-the-art architectures [44, 45] and data augmentation techniques [46]. Therefore, due to the prevalence of OOD data in safety critical applications, it is of the utmost importance that new algorithms be proposed which can improve the OOD performance of machine learning methods.

In this paper, we introduce a new framework for domain generalization which we call *Model-Based Domain Generalization* (MBDG). The key idea in our framework is to first learn transformations

that map data between domains and then to subsequently enforce invariance to these transformations. Under a general model of covariate shift and a novel notion of invariance to learned transformations, we use this framework to rigorously re-formulate the domain generalization problem as a semi-infinite constrained optimization problem. We then use this re-formulation to prove that a tight approximation of the domain generalization problem can be obtained by solving the empirical, parameterized dual for this semi-infinite problem. Finally, motivated by these theoretical insights, we propose a new algorithm for domain generalization; extensive experimental evidence shows that our algorithm advances the state-of-the-art on a range of benchmarks by up to thirty percentage points.

**Contributions.** Our contributions can be summarized as follows:

- We propose a new framework for domain generalization in which invariance is enforced to underlying transformations of data which capture inter-domain variation.
- Under a general model of covariate shift, we rigorously prove the equivalence of the domain generalization problem to a novel semi-infinite constrained statistical learning problem.
- We derive *data-dependent* duality gap bounds for the empirical parameterized dual of this semi-infinite problem, proving that tight approximations of the domain generalization problem can be obtained by solving this dual problem under the covariate shift assumption.
- We introduce a primal-dual style algorithm for domain generalization in which invariance is enforced over unsupervised generative models trained on data from the training domains.
- We empirically show that our algorithm significantly outperforms state-of-the-art baselines on several standard benchmarks, including `ColoredMNIST`, `Camelyon17-WILDS`, and `PACS`.

## 2 Related work

**Domain generalization.**    The rapid acceleration of domain generalization research has led to an abundance of principled algorithms, many of which distill knowledge from an array of disparate fields toward resolving OOD failure modes [47–50]. Among such works, one prominent thrust has been to learn predictors which have internal feature representations that are consistent across domains [51–62]. This approach is also popular in the field of unsupervised domain adaptation [63–67], wherein it is assumed that unlabeled data from the test domain is available during training [68–70]. Also related are works that seek to learn a kernel-based embedding of each domain in an underlying feature space [71, 72], and those that employ Model-Agnostic Meta Learning [73] to adapt to unseen domains [42, 74–81]. Recently, another prominent direction has been to design weight-sharing [82–85] and instance re-weighting schemes [86–88]. Unlike any of these approaches, we explicitly enforce hard invariance-based constraints on the underlying statistical domain generalization problem.

**Data augmentation.** Another approach to improve OOD performance is to augment the available training data. Among such methods, perhaps the most common is to leverage various forms of data augmentation [89–96]. Recently, several approaches have used style-transfer techniques and image-to-image translation networks [97–104] to augment the training domains with artificially-generated data [105–112]. Alternatively, rather than generating new data, [113–115] all remove textural features in the data to encourage domain invariance. Unlike the majority of these works, we do not perform data augmentation directly on the training objective; rather, we derive a principled primal-dual style algorithm which enforces invariance constraints on data generated by unsupervised generative models.

## 3 Domain generalization

The domain generalization setting is characterized by a pair of random variables $(X, Y)$ over instances $x \in \mathcal{X} \subseteq \mathbb{R}^d$ and corresponding labels $y \in \mathcal{Y}$, where $(X, Y)$ is jointly distributed according to an unknown probability distribution $\mathbb{P}(X, Y)$. Ultimately, the objective in this setting is to learn a predictor $f$ such that $f(X) \approx Y$, meaning that $f$ should be able to predict the labels $y$ of corresponding instances $x$ for each $(x, y) \sim \mathbb{P}(X, Y)$. However, unlike in standard supervised learning tasks, the domain generalization problem is complicated by the assumption that one cannot sample directly from $\mathbb{P}(X, Y)$. Rather, it is assumed that we can only measure $(X, Y)$ under different environmental conditions, each of which corrupts or varies the data in a different way. For example, in medical imaging tasks, these environmental conditions might correspond to the imaging techniques and stain patterns used at different hospitals; this is illustrated in Figure 1a.

To formalize this notion of environmental variation, we assume that data is drawn from a set of *environments* or *domains* $\mathcal{E}_{\text{all}}$ (see Figure 1b). Concretely, each domain $e \in \mathcal{E}_{\text{all}}$ can be identified with

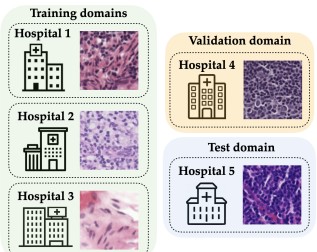
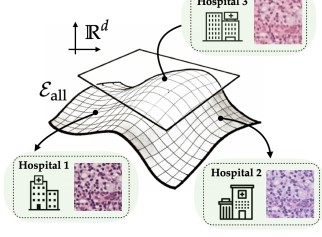
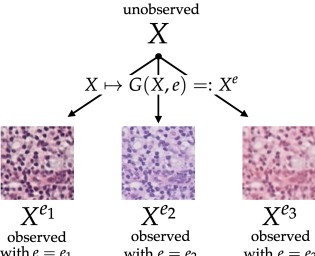

(a) In domain generalization, the data are drawn from a family of related domains. For example, in the `Camelyon17-WILDS` dataset [20], which contains images of cells, the domains correspond to different hospitals where these images were captured.

(b) Each data point in a domain generalization task is observed in a particular domain $e \in \mathcal{E}_{\text{all}}$. The set of all domains $\mathcal{E}_{\text{all}}$ can be thought of as an abstract space lying in $\mathbb{R}^p$. In `Camelyon17-WILDS`, this space $\mathcal{E}_{\text{all}}$ corresponds to the set of all possible hospitals.

(c) We assume that the variation from domain to domain is characterized by an underlying generative model $G(x, e)$, which transforms the unobserved random variable $X \mapsto G(X, e) := X^e$, where $X^e$ represents $X$ observed in any domain $e \in \mathcal{E}_{\text{all}}$.

Figure 1: An overview of the domain generalization problem setting used in this paper.

a pair of random variables $(X^e, Y^e)$, which together denote the observation of the random variable pair $(X, Y)$ in environment $e$. Given samples from a finite subset $\mathcal{E}_{\text{train}} \subsetneq \mathcal{E}_{\text{all}}$ of domains, the goal of the domain generalization problem is to learn a predictor $f$ that generalizes across all possible environments, implying that $f(X) \approx Y$. This can be summarized as follows:

**Problem 3.1** (Domain generalization). Let $\mathcal{E}_{\text{train}} \subsetneq \mathcal{E}_{\text{all}}$ be a finite subset of training domains, and assume that for each $e \in \mathcal{E}_{\text{train}}$, we have access to a dataset $\mathcal{D}^e := \{(x_j^e, y_j^e)\}_{j=1}^{n_e}$ sampled i.i.d. from $\mathbb{P}(X^e, Y^e)$. Given a function class $\mathcal{F}$ and a loss function $\ell : \mathcal{Y} \times \mathcal{Y} \to \mathbb{R}_{\geq 0}$, our goal is to learn a predictor $f \in \mathcal{F}$ using the data from the datasets $\mathcal{D}^e$ that minimizes the worst-case risk over the entire family of domains $\mathcal{E}_{\text{all}}$. That is, we want to solve the following optimization problem:

$$\underset{f \in \mathcal{F}}{\text{minimize}} \max_{e \in \mathcal{E}_{\text{all}}} \mathbb{E}_{\mathbb{P}(X^e, Y^e)} \, \ell(f(X^e), Y^e). \tag{DG}$$

In essence, in Problem 3.1 we seek a predictor $f \in \mathcal{F}$ that generalizes from the finite set of training domains $\mathcal{E}_{\text{train}}$ to perform well on the set of all domains $\mathcal{E}_{\text{all}}$. However, note that while the inner maximization in (DG) is over the set of all training domains $\mathcal{E}_{\text{all}}$, by assumption we do not have access to data from any of the domains $e \in \mathcal{E}_{\text{all}} \backslash \mathcal{E}_{\text{train}}$, making this problem challenging to solve. Indeed, as generalizing to arbitrary test domains is impossible [116], further structure is often assumed on the topology of $\mathcal{E}_{\text{all}}$ and on the corresponding distributions $\mathbb{P}(X^e, Y^e)$.

**Disentangling the sources of variation across environments.** The difficulty of a particular domain generalization task can be characterized by the extent to which the distribution of data in the unseen test domains $\mathcal{E}_{\text{all}} \backslash \mathcal{E}_{\text{train}}$ resembles the distribution of data in the training domains $\mathcal{E}_{\text{train}}$. For instance, if the domains are assumed to be convex combinations of the training domains, as is often the case in multi-source domain generalization [117–119], Problem 3.1 can be seen as an instance of distributionally robust optimization [120]. More generally, in a similar spirit to [116], we identify two forms of variation across domains: *covariate shift* and *concept shift*. These shifts characterize the extent to which the marginal distributions over instances $\mathbb{P}(X^e)$ and the instance-conditional distributions $\mathbb{P}(Y^e | X^e)$ differ between domains. We capture these shifts in the following definition:

**Definition 3.2** (Covariate shift & concept shift). Problem 3.1 is said to experience **covariate shift** if environmental variation is due to differences between the set of marginal distributions over instances $\{\mathbb{P}(X^e)\}_{e \in \mathcal{E}_{\text{all}}}$. On the other hand, Problem 3.1 is said to experience **concept shift** if environmental variation is due to changes amongst the instance-conditional distributions $\{\mathbb{P}(Y^e | X^e)\}_{e \in \mathcal{E}_{\text{all}}}$.

The growing domain generalization literature encompasses a great deal of past work, wherein both of these shifts have been studied in various contexts [121–125]. Indeed, as this literature has grown, new benchmarks have been developed which span the gamut between covariate and concept shift [126]. However, a large-scale empirical study recently showed that no existing algorithm can significantly outperform ERM across these standard domain generalization benchmarks when ERM is carefully

implemented [46]. As ERM is known to fail in the presence natural distribution shifts [127], this result highlights the critical need for new algorithms that can go beyond ERM toward solving Problem 3.1.

## 4   Model-based domain generalization

In what follows, we introduce a new framework for domain generalization that we call *Model-Based Domain Generalization* (MBDG). In particular, we prove that when Problem 3.1 is characterized solely by covariate shift, then under a natural invariance-based condition, Problem 3.1 is equivalent to an infinite-dimensional constrained statistical learning problem, which forms the basis of MBDG.

**Formal assumptions for MBDG.** In general, domain generalization tasks can be characterized by both covariate and concept shift. However, in this paper, we restrict the scope of our theoretical analysis to focus on problems in which inter-domain variation is due solely to covariate shift through an underlying model of data generation. Formally, we assume that the data in each domain $e \in \mathcal{E}_{\text{all}}$ is generated from the underlying random variable pair $(X, Y)$ via an unknown function $G$.

**Assumption 4.1.** Let $\delta_e$ denote a Dirac distribution for $e \in \mathcal{E}_{\text{all}}$. We assume that there exists[1] a measurable function $G : \mathcal{X} \times \mathcal{E}_{\text{all}} \to \mathcal{X}$, which we refer to as a *domain transformation model*, that parameterizes the inter-domain covariate shift via $\mathbb{P}(X^e) =^d G \# (\mathbb{P}(X) \times \delta_e) \ \forall e \in \mathcal{E}_{\text{all}}$, where $\#$ denotes the push-forward measure and $=^d$ denotes equality in distribution.

Informally, this assumption specifies that there should exist a function $G$ that relates the random variables $X$ and $X^e$ via $X \mapsto G(X, e) = X^e$. In past work, this particular setting in which the instances $X^e$ measured in an environment $e$ are related to the underlying random variable $X$ has been referred to as *domain shift* [128, §1.8]. In our medical imaging example, the domain shift captured by a domain transformation model would characterize the mapping from the underlying distribution $\mathbb{P}(X)$ over different cells to the distribution $\mathbb{P}(X^e)$ of images of these cells observed at a particular hospital; this is illustrated in Figure 1c, wherein inter-domain variation is due to varying colors and stain patterns encountered at different hospitals. On the other hand, in this example example, the label $y \sim Y$ describing whether a given cell contains a cancerous tumor should not depend on the lighting and stain patterns used at different hospitals. In this sense, while in other applications, e.g. the datasets introduced in [10], the instance-conditional distributions can vary across domains, in this paper we assume that inter-domain variation is *solely* characterized by the domain shift due to $G$.

**Assumption 4.2** (Domain shift). We assume that inter-domain variation is solely characterized by domain shift in the marginal distributions $\mathbb{P}(X^e)$, as described in Assumption 4.1. As a consequence, we assume that the instance-conditional distributions $\mathbb{P}(Y^e|X^e)$ are stable across domains, meaning that $Y^e$ and $Y$ are equivalent in distribution and that for each $x \in \mathcal{X}$ and $y \in \mathcal{Y}$, it holds that

$$\mathbb{P}(Y = y | X = x) = \mathbb{P}(Y^e = y | X^e = G(x, e)) \quad \forall e \in \mathcal{E}_{\text{all}}. \tag{1}$$

**Pulling back Problem 3.1.** The structure imposed on Problem 3.1 by Assumptions 4.1 and 4.2 provides a concrete way of parameterizing large families of distributional shifts in domain generalization problems. Indeed, the utility of these assumptions is that when taken together, they provide the basis for pulling-back Problem 3.1 onto the underlying distribution $\mathbb{P}(X, Y)$ via the domain transformation model $G$. This insight is captured in the following proposition:

**Proposition 4.3.** Under Assumptions 4.1 and 4.2, Problem 3.1 is equivalent to

$$\underset{f \in \mathcal{F}}{\text{minimize}} \ \underset{e \in \mathcal{E}_{\text{all}}}{\max} \ \mathbb{E}_{\mathbb{P}(X,Y)} \ \ell(f(G(X, e)), Y). \tag{2}$$

The proof of this fact is a straightforward consequence of the decomposition $\mathbb{P}(X^e, Y^e) = \mathbb{P}(Y^e|X^e) \cdot \mathbb{P}(X^e)$ in conjunction with Assumptions 4.1 and 4.2 (see Appendix C.2). Note that this result allows us to implicitly absorb each of the domain distributions $\mathbb{P}(X^e, Y^e)$ into the domain transformation model. Thus, the outer expectation in (2) is defined over the underlying distribution $\mathbb{P}(X, Y)$. On the other hand, just as in (DG), this problem is still a challenging statistical min-max problem. To this end, we next introduce a new notion of invariance with respect to domain transformation models, which allows us to reformulate the problem in (2) as a semi-infinite constrained optimization problem.

---

[1]Crucially, although we assume the existence of a domain transformation model $G$, we emphasize that for many problems, it may be impossible to obtain or derive a simple analytic expression for $G$. This topic will be discussed at length in Section 6 and in Appendix G.

**A new notion of model-based invariance.** Common to much of the domain generalization literature is the idea that predictors should be invariant to inter-domain changes. For instance, in [10] the authors seek to learn an *equipredictive representation* $\Phi : \mathcal{X} \to \mathcal{Z}$ [129], i.e. an intermediate representation that satisfies $\mathbb{P}(Y^{e_1}|\Phi(X^{e_1})) = \mathbb{P}(Y^{e_2}|\Phi(X^{e_2})) \quad \forall e_1, e_2 \in \mathcal{E}_{\text{all}}$. Despite compelling theoretical motivation for this approach, it has been shown that current algorithms which seek equipredictive representations do not significantly improve over ERM [130–133]. With this in mind and given the additional structure introduced in Assumptions 4.1 and 4.2, we introduce a new definition of invariance with respect to the variation captured by the underlying domain transformation model $G$.

**Definition 4.4** ($G$-invariance). Given a domain transformation model $G$, we say a classifier $f$ is **G-invariant** if it holds for all $e \in \mathcal{E}_{\text{all}}$ that $f(x) = f(G(x, e))$ almost surely when $x \sim \mathbb{P}(X)$.

Concretely, this definition says that a predictor $f$ is $G$-invariant if environmental changes under $G(x, e)$ cannot change the prediction returned by $f$. Intuitively, this notion of invariance couples with the definition of domain shift, in the sense that we expect that a prediction should return the same prediction for any realization of data under $G$. Thus, whereas equipredictive representations are designed to enforce invariance of in an intermediate representation space $\mathcal{Z}$, Definition 4.4 is designed to enforce invariance directly on the predictions made by $f$. In this way, in the setting of Figure 1, $G$-invariance would imply that the predictor $f$ would return the same label for a given cluster of cells regardless of the hospital at which these cells were imaged.

**Formulating the MBDG optimization problem.** The $G$-invariance property described in the previous section is the key toward reformulating the min-max problem in (2). Indeed, the following proposition follows from Assumptions 4.1 and 4.2 and from the definition of $G$-invariance.

**Proposition 4.5.** Under Assumptions 4.1 and 4.2, if we restrict the domain $\mathcal{F}$ of Problem 3.1 to the set of $G$-invariant predictors, then Problem 3.1 is equivalent to the following constrained problem:

$$P^\star \triangleq \underset{f \in \mathcal{F}}{\text{minimize}} \quad R(f) \triangleq \mathbb{E}_{\mathbb{P}(X,Y)} \ell(f(X), Y) \tag{MBDG}$$
$$\text{subject to} \quad f(x) = f(G(x, e)) \quad \text{a.e. } x \sim \mathbb{P}(X) \ \forall e \in \mathcal{E}_{\text{all}}.$$

Here a.e. stands for "almost everywhere" and $R(f)$ is the statistical risk of a predictor $f$ with respect to the underlying random variable pair $(X, Y)$. Note that unlike (2), (MBDG) is not a composite optimization problem, meaning that the inner maximization has been eliminated. In essence, the proof of Proposition 4.6 relies on the fact that $G$-invariance implies that predictions should not change across domains (see Appendix C.2). The optimization problem in (MBDG) forms the basis of our Model-Based Domain Generalization framework. To explicitly contrast this problem to Problem 3.1, we introduce the following concrete problem formulation for Model-Based Domain Generalization.

**Problem 4.6** (Model-Based Domain Generalization). As in Problem 3.1, let $\mathcal{E}_{\text{train}} \subsetneq \mathcal{E}_{\text{all}}$ be a finite subset of training domains and assume that we have access to datasets $\mathcal{D}^e \ \forall e \in \mathcal{E}_{\text{train}}$. Then under Assumptions 4.1 and 4.2, the goal of Model-Based Domain Generalization is to use the data from the training datasets to solve the semi-infinite constrained optimization problem in (MBDG).

Problem 4.6 offers a principled perspective on Problem 3.1 when data varies WRT an underlying domain transformation model. However, just as solving the min-max problem of Problem 3.1 is known to be difficult, the problem in (MBDG) is also challenging to solve for several reasons:

(C1) The $G$-invariance constraint in (MBDG) is strict and thus challenging to enforce.
(C2) Problem 4.6 is a constrained problem over an infinite-dimensional functional space $\mathcal{F}$.
(C3) We do access to the set of all domains $\mathcal{E}_{\text{all}}$ or to the underlying distribution $\mathbb{P}(X, Y)$.
(C4) We also generally do not have access to the underlying domain transformation model $G$.

In the ensuing sections, we explicitly address each of these challenges toward developing a tractable method for approximately solving Problem 4.6 with guarantees on optimality. In particular, we discuss challenges (C1), (C2), and (C3) in Section 5. We then discuss (C4) in Section F.

## 5 Data-dependent duality gap for MBDG

In this section, we offer a principled analysis of Problem 4.6. In particular, we first address (C1) by introducing a tight relaxation of the $G$-invariance constraint. Next, to resolve the fundamental

difficulty involved in solving constrained statistical problems highlighted in (C2), we formulate the parameterized dual problem, which is unconstrained and thus more suitable for learning with deep neural networks. Finally, to address (C3), we introduce an empirical version of the parameterized dual problem and explicitly characterize the data-dependent duality gap between this problem and Problem 4.6. At a high level, this analysis results in an *unconstrained* optimization problem which is guaranteed to produce a solution that is close to the solution of Problem 3.1 (see Theorem 5.3). In this section, we have chosen to present our results somewhat informally by deferring preliminary results, regularity assumptions, and proofs to the appendices.

**Addressing (C1) by relaxing the $G$-invariance constraint.** One of the most fundamental challenges in solving Problem 4.6 is the difficulty of enforcing the $G$-invariance equality constraint. To alleviate some of this difficulty, we introduce the following relaxation of Problem 4.6:

$$P^\star(\gamma) \triangleq \underset{f \in \mathcal{F}}{\text{minimize}} \; R(f) \quad \text{s.t.} \quad \mathcal{L}^e(f) \triangleq \mathbb{E}_{\mathbb{P}(X)} \, d\big(f(X), f(G(X, e))\big) \leq \gamma \quad \forall e \in \mathcal{E}_{\text{all}} \qquad (3)$$

where $\gamma > 0$ is a fixed margin the controls the extent to which we enforce $G$-invariance and $d : \mathcal{P}(\mathcal{Y}) \times \mathcal{P}(\mathcal{Y}) \to \mathbb{R}_{\geq 0}$ is a distance metric over the space of probability distributions on $\mathcal{Y}$. While at first glance this problem may appear to be a significant relaxation of the MBDG optimization problem in (MBDG), when $\gamma = 0$ and under mild conditions on $d$, the two problems are equivalent in the sense that $P^\star(0) = P^\star$ (see Proposition B.1). Indeed, we note that the conditions we require on $d$ are not restrictive, and include the KL-divergence and more generally the family of $f$-divergences. Moreover, when the margin $\gamma$ is strictly larger than zero, under the assumption that the perturbation function $P^\star(\gamma)$ is $L$-Lipschitz continuous, we show in Remark B.2 that $|P^\star - P^\star(\gamma)| \leq L\gamma$, meaning that the gap between the problems is relatively small when $\gamma$ is chosen to be small. In particular, when strong duality holds for (MBDG), this Lipschitz constant $L$ is equal to the $L^1$ norm of the optimal dual variable for (MBDG) (see Remark B.4).

**Addressing (C2) by formulating the parameterized dual problem.** As written, the relaxation in (3) is an infinite-dimensional constrained optimization problem over a functional space $\mathcal{F}$ (e.g. $L^2$ or the space of continuous functions). Optimization in this infinite-dimensional function space is not tractable, and thus we follow the standard convention by leveraging a finite-dimensional parameterization of $\mathcal{F}$, such as the class of deep neural networks [134, 135]. The approximation power of such a parameterization can be captured in the following definition:

**Definition 5.1** ($\epsilon$-parameterization). Let $\mathcal{H} \subseteq \mathbb{R}^p$ be a finite-dimensional parameter space. For $\epsilon > 0$, a function $\varphi : \mathcal{H} \times \mathcal{X} \to \mathcal{Y}$ is said to be an **$\epsilon$-parameterization** of $\mathcal{F}$ if it holds that for each $f \in \mathcal{F}$, there exists a parameter $\theta \in \mathcal{H}$ such that $\mathbb{E}_{\mathbb{P}(X)} \|\varphi(\theta, x) - f(x)\|_\infty \leq \epsilon$.

The benefit of using such a parameterization is that optimization is generally more tractable in the parameterized space $\mathcal{A}_\epsilon := \{\varphi(\theta, \cdot) : \theta \in \mathcal{H}\} \subseteq \mathcal{F}$. However, typical parameterizations often lead to nonconvex problems, wherein methods such as SGD cannot guarantee constraint satisfaction. And while several heuristic algorithms have been designed to enforce constraints over common parametric classes [136–141], these approaches cannot provide guarantees on the underlying statistical problem of interest [142]. Thus, to provide guarantees on the underlying statistical problem in Problem 4.6, given an $\epsilon$-parameterization $\varphi$ of $\mathcal{F}$, we consider the following saddle-point problem:

$$D_\epsilon^\star(\gamma) \triangleq \underset{\lambda \in \mathcal{P}(\mathcal{E}_{\text{all}})}{\text{maximize}} \; \underset{\theta \in \mathcal{H}}{\min} \; R(\theta) + \int_{\mathcal{E}_{\text{all}}} [\mathcal{L}^e(\theta) - \gamma] \, \mathrm{d}\lambda(e). \qquad (4)$$

where $\mathcal{P}(\mathcal{E}_{\text{all}})$ is the space of normalized probability distributions over $\mathcal{E}_{\text{all}}$ and $\lambda \in \mathcal{P}(\mathcal{E}_{\text{all}})$ is the (semi-infinite) dual variable. Here we have slightly abused notation to write $R(\theta) = R(\varphi(\theta, \cdot))$ and $\mathcal{L}^e(\theta) = \mathcal{L}^e(\varphi(\theta, \cdot))$. One can think of (4) as the dual problem to (3) solved over the parametric space $\mathcal{A}_\epsilon$. Notice that unlike Problem 4.6, the problem in (4) is *unconstrained*, making it much more amenable for optimization over the class of deep neural networks. Moreover, under mild conditions, the optimality gap between (3) and (4) can be explicitly bounded as follows:

**Proposition 5.2** (Parameterization gap). Let $\gamma > 0$ be given. Under mild regularity assumptions (see Assumption C.1 in Appendix C.3) on $\ell$ and $d$, there exists a small universal constant $k$ such that

$$P^\star(\gamma) \leq D_\epsilon^\star(\gamma) \leq P^\star(\gamma) + \epsilon k \left(1 + \left\|\lambda_{\text{pert}}^\star\right\|_{L^1}\right), \qquad (5)$$

where $\lambda_{\text{pert}}^\star$ is the optimal dual variable for a perturbed version of (3) in which the constraints are tightened to hold with margin $\gamma - k\epsilon$.

**Algorithm 1** Model-Based Domain Generalization (MBDG)

---

1: **Hyperparameters:** Primal step size $\eta_p > 0$, dual step size $\eta_d \geq 0$, margin $\gamma > 0$
2: **repeat**
3:     **for** minibatch $\{(x_j, y_j)\}_{j=1}^m$ in training dataset **do**
4:         $\tilde{x}_j \leftarrow \text{GENERATEIMAGE}(x_j) \; \forall j \in [m]$                 ▷ Generate model-based images
5:         $\text{distReg}(\theta) \leftarrow (1/m) \sum_{j=1}^m d(\varphi(\theta, x_j), \varphi(\theta, \tilde{x}_j))$        ▷ Calculate distance regularizer
6:         $\text{loss}(\theta) \leftarrow (1/m) \sum_{j=1}^m \ell(x_j, y_j; \varphi(\theta, \cdot))$            ▷ Calculate classification loss
7:         $\theta \leftarrow \theta - \eta_p \nabla_\theta [\, \text{loss}(\theta) + \lambda \cdot \text{distReg}(\theta) \,]$            ▷ Primal step for $\theta$
8:         $\lambda \leftarrow [\lambda + \eta_d (\text{distReg}(\theta) - \gamma)]_+$              ▷ Dual step for $\lambda$
9:     **end for**
10: **until** convergence
11:
12: **procedure** GENERATEIMAGE(x)
13:     Sample $e \sim \mathcal{N}(0, I)$                  ▷ $e$ is a latent code for MUNIT
14:     **return** $G(x, e)$                ▷ Return image produced by MUNIT
15: **end procedure**

---

In this way, solving the parameterized dual problem in (4) provides a solution that can be used to recover a close approximation of the primal problem in (3). To see this, observe that Prop. 5.2 implies that $|D_\epsilon^\star(\gamma) - P^\star(\gamma)| \leq \epsilon k (1 + ||\lambda_{\text{pert}}^\star||_{L^1})$. This tells us that the gap between $P^\star(\gamma)$ and $D_\epsilon^\star(\gamma)$ is small when we use a tight $\epsilon$-parameterization of $\mathcal{F}$.

**Addressing (C3) by bounding the empirical duality gap.** The parameterized dual problem in (4) gives us a principled way to address Problem 4.6 in the context of deep learning. However, complicating matters is the fact that we do not have access to the full distribution $\mathbb{P}(X, Y)$ or to data from any of the domains in $\mathcal{E}_{\text{all}} \backslash \mathcal{E}_{\text{train}}$. In practice, it is ubiquitous to solve optimization problems such as (4) over a finite sample of $N$ data points drawn from $\mathbb{P}(X, Y)$[2]. More specifically, given $\{(x_j, y_j)\}_{j=1}^N$ drawn i.i.d. according to $(X, Y)$, we consider the empirical counterpart of (4):

$$D_{\epsilon, N, \mathcal{E}_{\text{train}}}^\star(\gamma) \triangleq \underset{\lambda(e) \geq 0, \, e \in \mathcal{E}_{\text{train}}}{\text{maximize}} \; \min_{\theta \in \mathcal{H}} \hat{\Lambda}(\theta, \lambda) \triangleq \hat{R}(\theta) + \frac{1}{|\mathcal{E}_{\text{train}}|} \sum_{e \in \mathcal{E}_{\text{train}}} \left[ \hat{\mathcal{L}}^e(\theta) - \gamma \right] \lambda(e) \quad (6)$$

where $\hat{R}(\theta)$ and $\hat{\mathcal{L}}^e(\theta)$ are the empirical counterparts of $R(f)$ and $\mathcal{L}(f)$. Notably, the duality gap between the solution to (6) and (MBDG) can be explicitly bounded as follows.

**Theorem 5.3** (Data-dependent duality gap). Let $\epsilon > 0$ be given, and let $\varphi$ be an $\epsilon$-parameterization of $\mathcal{F}$. Under mild regularity assumptions on $\ell$ and $d$ and assuming that $\mathcal{A}_\epsilon$ has finite VC-dimension, with probability $1 - \delta$ over the $N$ samples from $\mathbb{P}(X, Y)$ we have that

$$|P^\star - D_{\epsilon, N, \mathcal{E}_{\text{train}}}^\star(\gamma)| \leq L\gamma + \epsilon k \left( 1 + \left\| \lambda_{\text{pert}}^\star \right\|_{L^1} \right) + \mathcal{O}\left( \sqrt{\log(N)/N} \right) \quad (7)$$

where $L$ is the Lipschitz constant of $P^\star(\gamma)$ and $k$ and $\lambda_{\text{pert}}^\star$ are as defined in Proposition 5.2.

The key message to take away from Theorem 5.3 is that given samples from $\mathbb{P}(X, Y)$, the duality gap incurred by solving the empirical problem in (6) is small when (a) the $G$-invariance margin $\gamma$ is small, (b) the parametric space $\mathcal{A}_\epsilon$ is a close approximation of $\mathcal{F}$, and (c) we have access to sufficiently many samples. Thus, assuming that Assumptions 4.1 and 4.2 hold, the solution to Problem 3.1 is closely-approximated by the solution to the empirical, parameterized dual problem in (6).

## 6 MBDG: A principled algorithm for domain generalization

Motivated by these theoretical insights, we now introduce a new domain generalization algorithm which is widely applicable to problems with or without covariate shift. Our algorithm consists of two steps. First, we learn an approximation of the underlying domain transformation model $G(x, e)$ using the data from the training domains $\mathcal{E}_{\text{train}}$. Next, we leverage $G$ toward solving the unconstrained dual optimization problem in (6) via a primal-dual iteration.

---

[2]Indeed, in practice we do not have access to any samples from $\mathbb{P}(X, Y)$. In Section 6, we argue that the $N$ samples from $\mathbb{P}(X, Y)$ can be replaced by the $\sum_{e \in \mathcal{E}_{\text{train}}} n_e$ samples drawn from the training datasets $\mathcal{D}^e$.

**Learning domain transformation models from data.** Regarding challenge (C4), critical to our approach is having access to the domain transformation model $G$. For the vast majority of settings, the underlying function $G(x, e)$ is not known a priori and cannot be represented by a simple expression. For example, obtaining a closed-form expression for a model that captures the variation in coloration, brightness, and contrast in the dataset shown in Figure 1 would be very challenging. While in general it is impossible to learn the true underlying domain transformation model when one only has access to data from the training domains, we argue that a realistic *approximation* of the underlying model can be learned from this data. To this end, to learn a domain transformation model, we train multimodal image-to-image translation networks on the training data. These networks are designed to transform samples from one dataset so that they resemble a diverse collection of images from another dataset. In particular, in each of the experiments in Section 7, we use the MUNIT architecture introduced in [102] to parameterize learned domain transformation models. As shown in Figure 5 and in Appendix G, models trained using the MUNIT architecture learn accurate and diverse transformations of the training data, which often generalize to generate images from new domains.

**Primal-dual iteration.** Given a learned approximation $G(x, e)$ of the underlying domain transformation model, the next step in our procedure is to use a primal-dual iteration [143] toward solving (6) using the datasets $\mathcal{D}^e$. We note that while our theory calls for data drawn from $\mathbb{P}(X, Y)$, the $G$-invariance condition implies that when (6) is feasible, $\varphi(\theta, x) \approx \varphi(\theta, x^e)$ when $x \sim \mathbb{P}(X)$, $x^e \sim \mathbb{P}^e(X)$, and $x^e = G(x, e)$. Therefore, the data from $\cup_{e \in \mathcal{E}_{\text{train}}} \mathcal{D}^e$ is a useful proxy for data drawn from $\mathbb{P}(X, Y)$. As the outer maximization in (6) is a linear program in $\lambda$, the primal-dual iteration can be characterized by alternating between the following steps:

$$\theta^{(t+1)} \in \rho\text{-}\operatorname*{argmin}_{\theta \in \mathcal{H}} \hat{\Lambda}(\theta, \lambda^{(t)}) \quad (8) \qquad \lambda^{(t+1)}(e) \leftarrow \left[\lambda^{(t)}(e) + \eta\left(\hat{\mathcal{L}}^e(\theta) - \gamma\right)\right]_+ \quad (9)$$

Here $[\cdot]_+ = \max\{0, \cdot\}$, $\eta$ is the dual step size, and $\rho$-$\operatorname{argmin}$ denotes a solution that is $\rho$-close to being a minimizer, i.e. we should that have $\hat{\Lambda}(\theta^{(t+1)}, \lambda^{(t)}) \leq \min_{\theta \in \mathcal{H}} \hat{\Lambda}(\theta, \lambda^{(t)}) + \rho$. We call (8) the primal step, and we call (9) the dual step. Furthermore, it can be shown that if this iteration is run for sufficiently many steps and with small enough step size, the iteration convergences with high probability to a solution which closely approximates the solution to Problem 4.6.

**Theorem 6.1** (Primal-dual convergence). Assuming that $\ell$ and $d$ are $[0, B]$-bounded, $\mathcal{H}$ has finite VC-dimension, and under mild regularity conditions on (6), the primal-dual pair $(\theta^{(T)}, \lambda^{(T)})$ obtained after running the alternating primal-dual iteration in (8) and (9) for $T$ steps with step size $\eta$, where

$$T = \left\lceil \frac{1}{2\eta\kappa} \right\rceil + 1 \qquad \text{and} \qquad \eta \leq \frac{2\kappa}{|\mathcal{E}_{\text{train}}| B^2} \qquad (10)$$

satisfies $|P^\star - \hat{\Lambda}(\theta^{(T)}, \mu^{(T)})| \leq K(\rho, \kappa, \epsilon) + \mathcal{O}(\sqrt{\log(N)/N})$. Here $\kappa$ is a constant that captures the regularity of (6) (see Appendix C.6) and $K(\rho, \kappa, \epsilon)$ is a small constant depending on $\rho$, $\kappa$, and $\epsilon$.

This means that by solving the empirical dual problem for sufficiently many steps, we can reach a solution that is close to solving the Model-Based Domain Generalization problem in Problem 4.6.

**Implementation of MBDG.** In practice, because (a) it may not be tractable to find a $\rho$-minimizer over $\mathcal{H}$ at each iteration and (b) there may be a large number of domains in $\mathcal{E}_{\text{train}}$, we propose two modifications of the primal-dual iteration in which we replace (8) with a stochastic gradient step and we use only one dual variable for all of the domains; we call this algorithm MBDG (see Algorithm 1). We provide results in Appendix E where one dual variable is used per training domain.

## 7 Experiments

We now evaluate the performance of MBDG on a range of standard domain generalization benchmarks. In the main text, we present results on `ColoredMNIST`, `Camelyon17-WILDS`, `FMoW-WILDS`, and `PACS`; we defer results on `VLCS` to the supplemental. For `ColoredMNIST`, `PACS`, and `VLCS`, we use the DomainBed package [46], facilitating comparison to a range of baselines. Model selection for each of these datasets was performed using hold-one-out cross-validation. For `Camelyon17-WILDS` and `FMoW-WILDS`, we use the repository provided with the WILDS dataset suite, and we perform model-selection using the out-of-distribution validation set provided in the WILDS repository. Further details concerning hyperparameter tuning and model selection are deferred to Appendix E.

Table 1: **ColoredMNIST.** We report accuracies for ColoredM-NIST. Model-selection was performed via cross-validation.

| Algorithm | +90% | +80% | -90% | Avg |
|---|---|---|---|---|
| ERM | $50.0 \pm 0.2$ | $50.1 \pm 0.2$ | $10.0 \pm 0.0$ | 36.7 |
| IRM | $46.7 \pm 2.4$ | $51.2 \pm 0.3$ | $23.1 \pm 10.7$ | 40.3 |
| GroupDRO | $50.1 \pm 0.5$ | $50.0 \pm 0.5$ | $10.2 \pm 0.1$ | 36.8 |
| Mixup | $36.6 \pm 10.9$ | $53.4 \pm 5.9$ | $10.2 \pm 0.1$ | 33.4 |
| MLDG | $50.1 \pm 0.6$ | $50.1 \pm 0.3$ | $10.0 \pm 0.1$ | 36.7 |
| CORAL | $49.5 \pm 0.0$ | $59.5 \pm 8.2$ | $10.2 \pm 0.1$ | 39.7 |
| MMD | $50.3 \pm 0.2$ | $50.0 \pm 0.4$ | $9.9 \pm 0.2$ | 36.8 |
| DANN | $49.9 \pm 0.1$ | $62.1 \pm 7.0$ | $10.0 \pm 0.1$ | 40.7 |
| CDANN | $63.2 \pm 10.1$ | $44.4 \pm 4.5$ | $9.9 \pm 0.2$ | 39.1 |
| MTL | $44.3 \pm 4.9$ | $50.7 \pm 0.0$ | $10.1 \pm 0.1$ | 35.0 |
| SagNet | $49.9 \pm 0.4$ | $49.7 \pm 0.3$ | $10.0 \pm 0.1$ | 36.5 |
| ARM | $50.0 \pm 0.3$ | $50.1 \pm 0.3$ | $10.2 \pm 0.0$ | 36.8 |
| VREx | $50.2 \pm 0.4$ | $50.5 \pm 0.5$ | $10.1 \pm 0.0$ | 36.9 |
| RSC | $49.6 \pm 0.3$ | $49.7 \pm 0.4$ | $10.1 \pm 0.0$ | 36.5 |
| MBDA | $72.0 \pm 0.1$ | $50.7 \pm 0.1$ | $22.5 \pm 0.0$ | 48.3 |
| MBDG-DA | $72.7 \pm 0.2$ | $71.4 \pm 0.1$ | $33.2 \pm 0.1$ | 59.0 |
| MBDG-Reg | $73.3 \pm 0.0$ | $73.7 \pm 0.0$ | $27.2 \pm 0.1$ | 58.1 |
| MBDG | $\mathbf{73.7 \pm 0.1}$ | $\mathbf{68.4 \pm 0.0}$ | $\mathbf{63.5 \pm 0.0}$ | $\mathbf{68.5}$ |

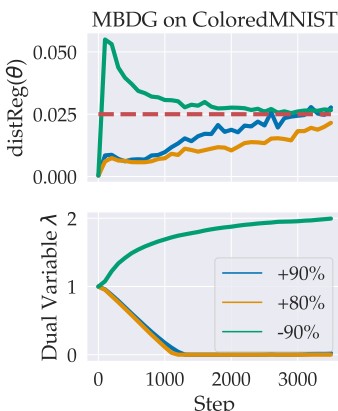

Figure 2: **Tracking dual variables.** We show the values of distReg($\theta$) and the dual variables $\lambda$ for the each MBDG models in Table 1. The margin $\gamma = 0.025$ is shown in red.

## 7.1 ColoredMNIST

We first consider the `ColoredMNIST` dataset [10], which is a standard domain generalization benchmark created by colorizing subsets of the MNIST dataset [144]. This dataset contains three domains, each of which is characterized by a different level of correlation between the label and digit color. As shown in Table 1, the MBDG algorithm improves over each baseline by nearly 30%. To understand the reasons behind this improvement, we consider three ablation studies on `ColoredMNIST`.

**Tracking the dual variables.** For the three MBDG classifiers in Table 1, we plot the regularization term distReg($\theta$) and the corresponding dual variable at each training step in Figure 2. Observe that for the +90% and +80% domains, the dual variables decay to zero, as the constraint is satisfied early on in training.

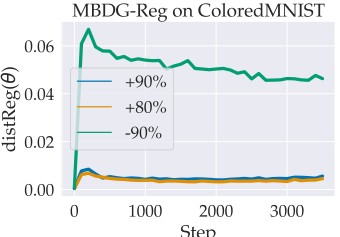

Figure 3: **Regularized MBDG.** We show the regularization value for each domain in `ColoredMNIST` for a fixed dual variable $\lambda = 1.0$.

On the other hand, the constraint for the -90% domain is not satisfied early on in training, and in response, the dual variable increases, gradually forcing constraint satisfaction. As we shown in the next subsection, without the dual update step, the constraints may never be satisfied (see Figure 3).

**Regularization vs. dual ascent.** A common trick for encouraging constraint satisfaction is to introduce soft constraints by adding a regularizer multiplied by a fixed multiplier to the objective. While this approach yields a related problem to (6) (see Appendix B.4) where the dual variables are fixed, there are few formal guarantees for this approach. Moreover, we show in Table 1 that when the dual variable is fixed during training (MBDG-Reg in Table 1), the performance drops significantly vis-a-vis MBDG. Notice that relative to Figure 2, the value of distReg($\theta$) is much larger than the margin, meaning that the constraint is not being satisfied.

**Ablation on data augmentation.** To study the efficacy of the MBDG algorithm, we consider two natural alternatives MBDG: (1) ERM with data augmentation through the learned model $G(x, e)$ (MBDA); and (2) MBDG with data augmentation through $G(x, e)$ on the training objective (MBDG-DA). As shown at the bottom of Table 1, while these variants significantly outperform the baselines, they not perform nearly as well as MBDG. Thus, while data augmentation can in some cases improve performance, the primal-dual iteration is a much more effective tool for enforcing invariance.

## 7.2 Camelyon17-WILDS and FMoW-WILDS

We next consider the `Camelyon17-WILDS` and `FMoW-WILDS` datasets from the WILDS family of domain generalization baselines [20]. Table 2 shows that on `Camelyon17-WILDS`, MBDG improves by more than 20 percentage points over the state-of-the-art baselines.

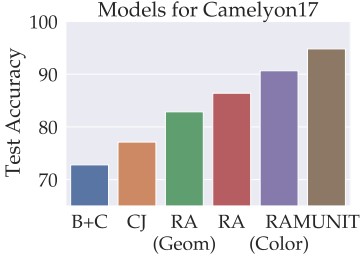

Figure 4: **Known vs. learned models $G(\mathbf{x}, \mathbf{e})$.** We compare the performance of MBDG for known models (first five columns) against a model that was trained with the data from the training domains using MUNIT.

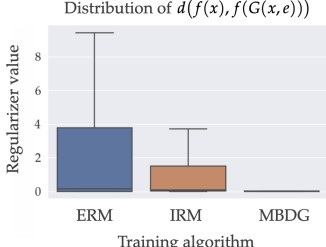

Figure 5: **Samples from learned models $G(\mathbf{x}, \mathbf{e})$.** We show samples from domain transformation models learned from the training data via the MUNIT architecture for `Camelyon17-WILDS`, `FMOW-WILDS`, and `PACS`.

Table 2: **WILDS datasets.** We report accuracies for `Camelyon17` and `FMoW`. For both datasets, we use the out-of-distribution validation set provided in the WILDS repository to perform model selection.

| Algorithm | Camelyon17-WILDS | FMoW-WILDS |
|---|---|---|
| ERM | $73.3 \pm 9.9$ | $51.3\ (0.4)$ |
| IRM | $60.9 \pm 15.3$ | $51.1\ (0.4)$ |
| ARM | $62.1 \pm 6.4$ | $47.9\ (0.3)$ |
| CORAL | $59.2 \pm 15.1$ | $49.6\ (0.5)$ |
| MBDG | $\mathbf{94.8 \pm 0.4}$ | $\mathbf{52.3 \pm 0.5}$ |

Figure 6: **Measuing invariance.** We measure the invariance of ERM, IRM, and MBDG to images generated by a model $G$ learned for `Camelyon17-WILDS`.

**Measuring $G$-invariance of trained classifiers.** In Section 4, we restricted our attention predictors satisfying the $G$-invariance condition. To test whether our algorithm successfully enforces $G$-invariance when $G$ is learned from data, we measure the distribution of $\text{distReg}(\theta)$ over all of the instances from the training domains of `Camelyon17-WILDS` for ERM, IRM, and MBDG. In Figure 6, observe that whereas MBDG is quite robust to changes under $G$, ERM and IRM are not nearly as robust. This property is key to the ability of MBDG to learn invariant representations across domains.

**Ablation on learning models vs. data augmentation.** Rather than learning $G$ from data, a heuristic alternative is to replace the GENERATEIMAGE procedure in Algorithm 1 with standard data augmentation transformations. In Figure 4, we investigate this approach with five different forms of data augmentation: B+C (brightness and contrast), CJ (color jitter), and three variants of RandAugment [145] (RA, RA-Geom, and RA-Color). More details concerning these data augmentation schemes are given in Appendix E. The bars in Figure 4 show that although these schemes offer strong performance in our MBDG framework, the learned model trained using MUNIT performs best.

### 7.3 PACS

In this subsection, we highlight selected results for the `PACS` dataset. Due to spatial limitations, we report the top-performing baselines in the main text, and defer the full set of results to Appendix E. Notably, MBDG beats the current SOTA by nearly 2% when averaged over the four domains. Of note in Table 3 is the result on the "sketch" (S) subset, wherein MBDG improves by more than 10% over the baselines.

Table 3: **PACS.** We report classification accuracies for `PACS`. Model-selection was performed via cross-validation.

| Algorithm | A | C | P | S | Avg |
|---|---|---|---|---|---|
| ERM | $83.2 \pm 1.3$ | $76.8 \pm 1.7$ | $\mathbf{97.2 \pm 0.3}$ | $74.8 \pm 1.3$ | $83.0$ |
| MTL | $\mathbf{85.6 \pm 1.5}$ | $78.9 \pm 0.6$ | $97.1 \pm 0.3$ | $73.1 \pm 2.7$ | $83.7$ |
| RSC | $83.7 \pm 1.7$ | $\mathbf{82.9 \pm 1.1}$ | $95.6 \pm 0.7$ | $68.1 \pm 1.5$ | $82.6$ |
| MBDG | $80.6 \pm 1.1$ | $79.3 \pm 0.2$ | $97.0 \pm 0.4$ | $\mathbf{85.2 \pm 0.2}$ | $\mathbf{85.6}$ |

# 8 Acknowledgements and disclosure of funding

The authors would like to thank Samuel Sokota for his helpful feedback.

The research of Alexander Robey and Hamed Hassani is supported by NSF Grants 1837253, 1943064, 1934876, AFOSR Grant FA9550-20-1-0111, and DCIST-CRA. Alexander Robey and George J. Pappas are supported by the NSF-Simons Program on the Mathematics of Deep Learning, ARL CRA DCIST W911NF-17-2-0181, and NSF 2038873.

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
