 



(a) **Domain shift.** In this paper, we assume that the instances $X^e$ from a domain $e \in \mathcal{E}_{\text{all}}$ are generated by a domain transformation model $G(X, e)$, resulting in domain shift. Thus, in the above SCM, $X$ and $e$ are the sole causal ancestors of $X^e$. Further, we assume that $e$ is not a causally related to $Y = Y^e$.

(b) **Concept shift.** In this figure, we illustrate a causal data generating model in which the instances $X^e$ can be (spuriously) correlated with the label $Y$, leading to concept shift. Note that unlike in the SCM shown in (a), in this SCM, $Y$ is also a causal parent of $X^e$.

Figure 7: **Causal interpretations of domain generalization tasks.** We compare structural causal models (SCMs) for covariate shift and concept shift. Throughout, the environment $e \in \mathcal{E}_{\text{all}}$ is assumed to be independent of $(X, Y)$, i.e. $e \perp\!\!\!\perp (X, Y)$.

## A  A causal interpretation of MBDG

The language of causal inference provides further intuition for the structure imposed on Problem 3.1 by Assumptions 4.1 and 4.2. In particular, the structural causal model (SCM) for problems in which data is generated according to the mechanism described in Assumptions 4.1 and 4.2 is shown in Figure 7a. Recall that in Assumption 4.1 imposes that $X$ and $e$ are *causes* of the random variable $X^e$ via the mechanism $X^e = G(X, e)$. This results in the causal links $e \longrightarrow X^e \longleftarrow X$. Further, in Assumption 4.2, we assume that $\mathbb{P}(Y^e|X^e)$ is fixed across environments, meaning that the label $Y$ is independent of the environment $e$. In Figure 7a, this translates to there being no causal link between $e$ and $Y$.

To offer a point of comparison, in Figure 7b, we show a different SCM that does not fulfill our assumptions. Notice that in this SCM, $Y$ and $e$ are both causes of $X^e$, meaning that the distributions $\mathbb{P}(Y^e|X^e)$ can vary in domain dependent ways. This gives rise to concept shift, which has also been referred to as *spurious correlation* [10]. Notably, the SCM shown in Figure 7b corresponds to the data generating procedure used to construct the `ColoredMNIST` dataset [10], wherein the MNIST digits in various domains $X^e$ are (spuriously) colorized according to the label $Y$.[3]

---

[3]While the data-generating mechanism for `ColoredMNIST` does not fulfill our assumptions, the algorithm we propose in Section 6 still empirically achieves state-of-the-art results on `ColoredMNIST`.

# B Further theoretical results and discussion

## B.1 On the optimality of relaxation of Problem 4.6 in (3)

In Section 5 of the main text, we claimed that the relaxation introduced in (3) was tight under certain conditions. In this section of the appendix, we formally enumerate the conditions under which the relaxation is tight. Further, we show that the tightness of the relaxation can be characterized by the margin parameter $\gamma$.

### B.1.1 The case when $\gamma = 0$

In Section 5, we claimed that the relaxation of the Model-Based Domain Generalization problem given in (3) was tight when $\gamma = 0$ under mild conditions on the distance metric $d$. In particular, we simply require that $d(\mathbb{P}, \mathbb{T}) = 0$ if and only if $\mathbb{P} = \mathbb{T}$ almost surely. We emphasize that this condition is not overly restrictive. Indeed, a variety of distance metrics, including the KL-divergence and more generally the family of $f$-divergences, satisfy this property (c.f. [146, Theorem 8.6.1]). In what follows, we formally state and prove this result.

**Proposition B.1.** Let $d$ be a distance metric between probability measures for which it holds that $d(\mathbb{P}, \mathbb{T}) = 0$ for two distributions $\mathbb{P}$ and $\mathbb{T}$ if and only if $\mathbb{P} = \mathbb{T}$ almost surely. Then $P^\star(0) = P^\star$.

*Proof.* The idea in this proof is simply to leverage the fact a non-negative random variable has expectation zero if and only if it is zero almost everywhere. For ease of exposition, we remind the reader of the definition of the relaxed constraints: $\mathcal{L}^e(f) := \mathbb{E}_{\mathbb{P}(X)} d(f(X), f(G(X, e)))$.

First, observe that because $d(\cdot, \cdot)$ is a metric, it is non-negative-valued. Then the following statement is trivial

$$\mathcal{L}^e(f) \leq 0 \iff \mathcal{L}^e(f) = 0. \tag{11}$$

Next, we claim that under the assumptions given in the statement of the proposition, $\mathcal{L}^e(f) = 0$ is equivalent to the $G$-invariance condition. To verify this claim, for simplicity we start by defining the random variable

$$Z_e \triangleq d\big(f(X), f(G(X, e))\big) \tag{12}$$

and note that by construction $Z_e \geq 0$ a.e. and $\mathcal{L}^e(f) = \mathbb{E}_{\mathbb{P}(X)} Z_e$. Now consider that because $Z_e$ is non-negative and has an expectation of zero, we have that $\mathbb{E}_{\mathbb{P}(X)} Z_e = 0$ if and only if $Z_e = 0$ almost surely (c.f. Prop. 8.1 in [147]). In other words, we have shown that

$$\mathcal{L}^e(f) = 0 \iff d\big(f(x), f(G(x, e))\big) = 0 \quad \text{a.e. } x \sim \mathbb{P}(X) \tag{13}$$

holds for each $e \in \mathcal{E}_{\text{all}}$. Now by assumption, we have that for any two distributions $\mathbb{P}$ and $\mathbb{T}$ sharing the same support that $d(\mathbb{P}, \mathbb{T}) = 0$ holds if and only if $\mathbb{P} = \mathbb{T}$ almost surely. Applying this to (13), we have that

$$\mathcal{L}^e(f) = 0 \iff f(x) = f(G(x, e)) \quad \text{a.e. } x \sim \mathbb{P}(X). \tag{14}$$

Altogether we have shown that $\mathcal{L}^e(f) \leq 0$ if and only if $f$ is $G$-invariant. Thus, when $\gamma = 0$, the optimization problems in (MBDG) and (3) are equivalent, which implies that $P^\star(0) = P^\star$. □

## B.2 The case when $\gamma > 0$

When $\gamma > 0$, the relaxation is no longer tight. However, if the perturbation function $P^\star(\gamma)$ is assumed to be Lipschitz continuous, we can directly characterize the tightness of the bound.

**Remark B.2.** Let us assume that the perturbation function $P^\star(\gamma)$ is $L$-Lipschitz continuous in $\gamma$. Then given Proposition B.1, it follows that $|P^\star - P^\star(\gamma)| \leq L\gamma$.

*Proof.* Observe that by Proposition B.1, we have that $P^\star = P^\star(0)$. It follows that

$$|P^\star - P^\star(\gamma)| = |P^\star(0) - P^\star(\gamma)| \tag{15}$$
$$\leq L|0 - \gamma| \tag{16}$$
$$= L\gamma \tag{17}$$

where the inequality in (16) follows by the definition of Lipschitz continuity. □

We note that in general the perturbation function $P^\star(\gamma)$ cannot be guaranteed to be Lipschitz. However, as we will show in Remark B.4, when strong duality holds for (MBDG), $P^\star(\gamma)$ turns out to be Lipschitz continuous with a Lipschitz constant equal to the $L^1$ norm of optimal dual variable for the dual problem to (MBDG). Before proving this result, we state a preliminary lemma from [148].

**Lemma B.3** (§5.6.2 in [148]). Consider a generic optimization problem

$$p^\star \triangleq \min_{x \in \mathbb{R}^d} f_0(x) \quad \text{subject to} \quad f_i(x) \leq 0 \quad \forall i \{1, \ldots, m\}. \tag{18}$$

Assume that strong duality holds for this problem, and let $\lambda^\star$ denote an optimal dual variable. Define the perturbation function as follows:

$$p^\star(u) \triangleq \min_{x \in \mathbb{R}^d} f_0(x) \quad \text{subject to} \quad f_i(x) \leq u_i \quad \forall i \in \{1, \ldots, m\} \tag{19}$$

where $u \in \mathbb{R}^m$. Then it holds that $p^\star(u) \geq p^\star - u^\top \lambda^\star$.

This useful result, which follows from a simple one-line proof in §5.6.2 of [148], shows that the perturbation function $p^\star(u)$ can be related to the optimal value of the unperturbed problem via the optimal dual variable. We can readily use a semi-infinite version of this lemma to prove the following remark:

**Remark B.4.** Consider the dual problem to (MBDG):

$$D^\star \triangleq \max_{\lambda \in \mathcal{B}(\mathcal{E}_{\text{all}})} \min_{f \in \mathcal{F}} R(f) + \int_{\mathcal{E}_{\text{all}}} [L^e(f) - \gamma] \, \mathrm{d}\nu(e) \tag{20}$$

where $\mathcal{B}(\cdot)$ denotes the cone of non-regular, non-negative Borel measures supported on its argument [149]. Assume that strong duality holds, and let $\nu^\star$ denote an optimal dual variable for this problem. Then it holds that

$$|P^\star - P^\star(\gamma)| \leq \gamma \|\nu^\star\|_{L^1}. \tag{21}$$

*Proof.* The idea here is to apply Lemma B.3 for the constant function defined by $u = u(e) = \gamma$ $\forall e \in \mathcal{E}_{\text{all}}$. To begin, let $\langle \cdot, \cdot \rangle$ denote the standard inner product on $L^2$; i.e. $\langle f, g \rangle = \int_{\mathcal{E}_{\text{all}}} f(e)g(e)\mathrm{d}e$ for $f, g \in L^2(\mathcal{E}_{\text{all}})$. In this way, we find that

$$P^\star - \langle u, \nu^\star \rangle \leq P^\star(\gamma) \leq P^\star \tag{22}$$

where the second inequality holds because for $\gamma$ strictly larger than zero, the relaxation in (3) corresponds to an expansion of the feasible set of relative to (MBDG). In this case, since $u$ is constant, a simple calculation shows that

$$\langle u, \nu^\star \rangle = \int_{\mathcal{E}_{\text{all}}} \nu^\star(e)u(e)\mathrm{d}e = \gamma \int_{\mathcal{E}_{\text{all}}} \nu^\star(e)\mathrm{d}e = \gamma \cdot \|\nu^\star\|_{L^1} \tag{23}$$

where in the last step we have used the fact that the optimal dual variable $\nu^\star \succeq 0$. Now if we apply this result to (22), we find that

$$P^\star - \gamma \|\nu^\star\| \leq P^\star(\gamma) \leq P^\star, \tag{24}$$

which directly implies the desired result. $\qquad\square$

## B.3 Relationship to constrained PAC learning

Recently, the authors of [150] introduced the Probably Approximately Correct Constrained (PACC) framework, which extends the classical PAC framework to constrained problems. In particular, recall the following definition of agnostic PAC learnability:

**Definition B.5** (PAC learnability). A hypothesis class $\mathcal{H}$ is said to be (agnostic) PAC learnable if for every $\epsilon, \delta \in (0, 1)$ and every distribution $\mathbb{P}_0$, there exists a $\theta^\star \in \mathcal{H}$ which can be obtained from $N \geq N_{\mathcal{H}}(\epsilon, \delta)$ samples from $\mathbb{P}_0$ such that $\mathbb{E}\,\ell(\varphi(\theta, X), Y) \leq U^\star + \epsilon$ with probability $1 - \delta$, where

$$U^\star \triangleq \underset{\theta \in \mathcal{H}}{\text{minimize}} \; \mathbb{E}_{\mathbb{P}_0(X,Y)} \ell(\varphi(\theta, X), Y) \tag{25}$$

The authors of [150] extended this definition toward studying the learning theoretic properties of constrained optimization problems of the form

$$C^\star \triangleq \underset{\theta \in \mathcal{H}}{\text{minimize}} \quad \mathbb{E}_{\mathbb{P}_0(X,Y)} \, \ell_0(\varphi(\theta, X), Y) \tag{26}$$

$$\text{subject to} \quad \mathbb{E}_{\mathbb{P}_i(X,Y)} \, \ell_i(\varphi(\theta, X), Y) \leq c_i \quad \text{for } i \in \{1, \ldots, m\} \tag{27}$$

$$\ell_j(\varphi(\theta, X), Y) \leq c_j \ \mathbb{P}_j - \text{a.e.} \quad \text{for } j \in \{m+1, \ldots m+q\} \tag{28}$$

via the following definition:

**Definition B.6** (PACC learnability). A hypothesis class $\mathcal{H}$ is said to be PACC learnable if for every $\epsilon, \delta \in (0, 1)$ and every distribution $\mathcal{P}_i$ for $i \in \{0, \ldots, m+q\}$, there exists a $\theta^\star \in \mathcal{H}$ which can be obtained from $N \geq N_{\mathcal{H}}(\epsilon, \delta)$ samples from each of the distributions $\mathbb{P}_i$ such that, with probability $1 - \delta$, $\theta^\star$ is:

(1) approximately optimal, meaning that

$$\mathbb{E}_{\mathbb{P}_0} \, \ell_0(\varphi(\theta^\star, X), Y) \leq C^\star + \epsilon \tag{29}$$

(2) approximately feasible, meaning that

$$\mathbb{E}_{\mathbb{P}_i(X,Y)} \, \ell_i(\varphi(\theta, X), Y) \leq c_i + \epsilon \quad \text{for } i \in \{1, \ldots, m\} \tag{30}$$

$$\ell_j(\varphi(X), Y) \leq c_j \ \forall (x, y) \in \mathcal{K}_j \quad \text{for } j \in \{m+1, \ldots, m+q\} \tag{31}$$

where $\mathcal{K}_j \subseteq \mathcal{X} \times \mathcal{Y}$ are sets of $\mathbb{P}_j$ measure at least $1 - \epsilon$.

One of the main results in [150] is that a hypothesis class $\mathcal{H}$ is PAC learnable if and only if it is PACC learnable.

Now if we consider the optimization problem in (26), we see that the admissible constraints are both inequality constraints. In contrast, the optimization problem in Problem 4.6 contains a family of equality constraints. Thus, in addition to easing the burden of enforcing hard $G$-invariance, the relaxation in (3) serves to manipulate the Model-Based Domain Generalization problem into a form compatible with (26). This is one of the key steps that sets the stage for deriving the learning theoretic guarantees for Model-Based Domain Generalization (e.g. Theorems 5.3 and 6.1).

## B.4 Regularization vs. dual ascent

A common trick for encouraging constraint satisfaction is to introduce soft constraints by adding a regularizer multiplied by a fixed penalty weight to the objective. As noted in Section 7, this approach yields a similar optimization problem to (6). In particular, the regularized version of (6) is the following:

$$\hat{D}^\star_{\epsilon, N, \mathcal{E}_{\text{train}}} \triangleq \underset{\theta \in \mathcal{H}}{\text{minimize}} \, \hat{R}(\theta) + \frac{1}{|\mathcal{E}_{\text{train}}|} \sum_{e \in \mathcal{E}_{\text{train}}} \left[ \hat{\mathcal{L}}^e(\theta) - \gamma \right] w(e) \tag{32}$$

where $w(e) \geq 0 \ e \in \mathcal{E}_{\text{train}}$ are weights that are chosen as hyperparameters. From an optimization perspective, the benefit of such an objective is that gradient-based algorithms are known to converge to local minima given small enough step sizes (MBDG). However, classical results in learning theory can only provide generalization guarantees on the aggregated objective, rather than on each term individually. Furthermore, the choice of the penalty weights $w(e)$ is non-trivial and often requires significant domain knowledge, limiting the applicability of this approach.

In contrast, in primal-dual style algorithms, the weights $\lambda(e)$ are not fixed beforehand. Rather, the $\lambda(e)$ are updated iteratively via the dual ascent step described in line 8 of Algorithm 1. Furthermore, as we showed in the main text, the optimal value of the primal problem $P^\star$ can be directly related to the solution of the empirical dual problem in (6) via Theorem 5.3. Such guarantees are not possible in the regularization case, which underscores the benefits of the primal-dual iteration over the more standard regularization approach.

## C  Omitted proofs

In this appendix, we provide the proofs that were omitted in the main text. For ease of exposition, we restate each result before proving it so that the reader can avoid scrolling back and forth between the main text and the appendices.

### C.1  Proof of Proposition 4.3

**Proposition 4.3.** Under Assumptions 4.1 and 4.2, Problem 3.1 is equivalent to

$$\underset{f \in \mathcal{F}}{\text{minimize}} \ \max_{e \in \mathcal{E}_{\text{all}}} \ \mathbb{E}_{\mathbb{P}(X,Y)} \, \ell(f(G(X,e)), Y). \tag{33}$$

*Proof.* The main idea in this proof is the following. First, we decompose the joint distribution $\mathbb{P}(X^e, Y^e)$ into $\mathbb{P}(Y^e | X^e) \cdot \mathbb{P}(X^e)$ to expand the risk term in the objective of (DG). Next, we leverage Assumptions 4.1 and 4.2 to rewrite the outer and inner expectations engendered by the tower property. Finally, we undo our expansion to arrive at at the statement of the proposition.

To begin, observe that by the our decomposition $\mathbb{P}(X^e, Y^e) = \mathbb{P}(Y^e | X^e) \cdot \mathbb{P}(X^e)$ of the joint expectation, we can rewrite the objective of (DG) in the following way:

$$\mathbb{E}_{\mathbb{P}(X^e, Y^e)} \, \ell(f(X^e), Y^e) = \mathbb{E}_{\mathbb{P}(X^e)} \left[ \mathbb{E}_{\mathbb{P}(Y^e | X^e)} \, \ell(f(X^e), Y^e) \right]. \tag{34}$$

Then, recall that by Assumption 4.2, we have that $\mathbb{P}(Y^e | X^e) = \mathbb{P}(Y | X) \, \forall e \in \mathcal{E}_{\text{all}}$, i.e. the conditional distribution of labels given instances is the same across domains. Thus, if we consider the inner expectation in (34), it follows that

$$\mathbb{E}_{\mathbb{P}(Y^e | X^e)} \, \ell(f(X^e), Y^e) = \mathbb{E}_{\mathbb{P}(Y | X)} \, \ell(f(X), Y). \tag{35}$$

Now observe that under Assumption 4.1, we have that $\mathbb{P}(X^e) \overset{d}{=} G \# (\mathbb{P}(X), \delta_e)$. Therefore, a simple manipulation reveals that

$$\mathbb{E}_{\mathbb{P}(X^e)} \left[ \mathbb{E}_{\mathbb{P}(Y^e | X^e)} \, \ell(f(X), Y) \right] = \mathbb{E}_{G \# (\mathbb{P}(X), \delta_e)} \left[ \mathbb{E}_{\mathbb{P}(Y | X)} \, \ell(f(X), Y) \right] \tag{36}$$

$$= \mathbb{E}_{\mathbb{P}(X)} \left[ \mathbb{E}_{\mathbb{P}(Y | X)} \, \ell(f(G(X, e)), Y) \right] \tag{37}$$

$$= \mathbb{E}_{\mathbb{P}(X, Y)} \, \ell(f(G(X, e)), Y), \tag{38}$$

where the final step again follows from the tower property of expectation. Therefore, by combining (34) and (38), we conclude that

$$\mathbb{E}_{\mathbb{P}(X^e, Y^e)} \, \ell(f(X^e), Y^e) = \mathbb{E}_{\mathbb{P}(X, Y)} \, \ell(f(G(X, e)), Y), \tag{39}$$

which directly implies the statement of the proposition. □

### C.2  Proof of Proposition 4.5

**Proposition 4.5.** Under Assumptions 4.1 and 4.2, if we restrict the feasible set to the set of $G$-invariant predictors, then Problem 3.1 is equivalent to the following semi-infinite constrained problem:

$$P^\star \triangleq \underset{f \in \mathcal{F}}{\text{minimize}} \quad R(f) \triangleq \mathbb{E}_{\mathbb{P}(X,Y)} \, \ell(f(X), Y) \tag{40}$$

$$\text{subject to} \quad f(x) = f(G(x, e)) \quad \text{a.e. } x \sim \mathbb{P}(X) \ \forall e \in \mathcal{E}_{\text{all}}.$$

*Proof.* The main idea in this proof is simply to leverage the definition of $G$-invariance and the result of Prop. 4.3. Starting from Prop. 4.3, we see that by restricting the feasible set to the set of $G$ invariant predictors, the optimization problem in (2) can be written as

$$P^\star = \underset{f \in \mathcal{F}}{\text{minimize}} \quad \max_{e \in \mathcal{E}_{\text{all}}} \ \mathbb{E}_{\mathbb{P}(X,Y)} \, \ell(f(G(X, e)), Y) \tag{41}$$

$$\text{subject to} \quad f(x) = f(G(x, e)) \quad \text{a.e.} x \sim \mathbb{P}(X), \ \forall e \in \mathcal{E}_{\text{all}} \tag{42}$$

Now observe that due to the constraint, we can replace the $f(G(X, e))$ term in the objective with $f(X)$. Thus, the above problem is equivalent to

$$P^\star = \underset{f \in \mathcal{F}}{\text{minimize}} \quad \max_{e \in \mathcal{E}_{\text{all}}} \ \mathbb{E}_{\mathbb{P}(X,Y)} \, \ell(f(X), Y) \tag{43}$$

$$\text{subject to} \quad f(x) = f(G(x, e)) \quad \text{a.e. } x \sim \mathbb{P}(X), \ \forall e \in \mathcal{E}_{\text{all}} \tag{44}$$

Now observe that the objective in (43) is free of the optimization variable $e \in \mathcal{E}_{\text{all}}$. Therefore, we can eliminate the inner maximization step in (43), which verifies the claim of the proposition. □

## C.3 Proof of Proposition 5.2

Before proving Proposition 5.2, we formally state the assumptions we require on $\ell$ and $d$. These assumptions are enumerated in the following Assumption:

**Assumption C.1.** We make the following assumptions:

1. The loss function $\ell$ is non-negative, convex, and $L_\ell$-Lipschitz continuous in it's first argument, i.e.

$$|\ell(f_1(x), y) - \ell(f_2(x), y)| \leq \|f_1(x) - f_2(x)\|_\infty \tag{45}$$

2. The distance metric $d$ is non-negative, convex, and satisfies the following uniform Lipschitz-like inequality for some constant $L_d > 0$:

$$|d(f_1(x), f_1(G(x, e))) - d(f_2(x), f_2(G(x, e)))| \leq L_d \|f_1(x) - f_2(x)\|_\infty \quad \forall e \in \mathcal{E}_{\text{all}}. \tag{46}$$

3. There exists a predictor $f \in \mathcal{F}$ such that $\mathcal{L}^e(f) < \gamma - \epsilon \cdot \max\{L_\ell, L_d\} \; \forall e \in \mathcal{E}_{\text{all}}$.

At a high level, these assumptions necessitate that $\ell$ and $d$ are sufficiently regular and that the problem is strictly feasible with a particular margin $\epsilon \cdot \max\{L_\ell, L_d\}$. In particular, this final assumption is essential as it implies that strong duality holds for (3), which is a key technical element of the proof. Given these assumptions, we restate Proposition 5.2 below:

**Proposition 5.2.** Let $\gamma > 0$ be given. Then under Assumption C.1, it holds that

$$P^\star(\gamma) \leq D_\epsilon^\star(\gamma) \leq P^\star(\gamma) + \epsilon \left(1 + \left\|\lambda_{\text{pert}}^\star\right\|_{L^1}\right) \cdot \max\{L_\ell, L_d\} \tag{47}$$

where $\lambda_{\text{pert}}^\star$ is the optimal dual variable for a perturbed version of (3) in which the constraints are tightened to hold with margin $\gamma - \epsilon \cdot \max\{L_\ell, L_d\}$. In particular, this result implies that

$$|P^\star(\gamma) - D_\epsilon^\star(\gamma)| \leq \epsilon \left(1 + \left\|\lambda_{\text{pert}}^\star\right\|_{L^1}\right) \cdot \max\{L_\ell, L_d\} \tag{48}$$

*Proof.* In this proof, we extend the results of [142] to optimization problems with an infinite number of constraints. The key insight toward deriving the lower bound is to use the fact that maximizing over the $\epsilon$-parameterization of $\mathcal{F}$ yields a sub-optimal result vis-a-vis maximizing over $\mathcal{F}$. On the other hand, the upper bound, which requires slightly more machinery, leverages Jensen's and Hölder's inequalities along with the definition of the $\epsilon$-parameterization to over-approximate the parameter space via a Lipschitz $\epsilon$-ball covering argument.

**Step 1.** In the first step, we prove the lower bound in (47). To begin, we define the dual problem to the relaxed Model-Based Domain Generalization problem in (3) in the following way:

$$D^\star(\gamma) \triangleq \underset{\lambda \in \mathcal{B}(\mathcal{E}_{\text{all}})}{\text{maximize}} \; \underset{f \in \mathcal{F}}{\min} \; \Lambda(f, \lambda) \triangleq R(f) + \int_{\mathcal{E}_{\text{all}}} [\mathcal{L}^e(\varphi(\theta, \cdot)) - \gamma] \, d\lambda(e). \tag{49}$$

where with a slight abuse of notation, we redefine the Lagrangian $\Lambda$ from (4) in its first argument. Now recall that by assumption, there exists a predictor $f \in \mathcal{F}$ such that $\mathcal{L}(f) < \gamma \; \forall e \in \mathcal{E}_{\text{all}}$. Thus, Slater's condition holds [148], and therefore so too does strong duality. Now let $f^\star$ be optimal for the primal problem (3), and let $\lambda^\star \in \mathcal{B}(\mathcal{E}_{\text{all}})$ be dual optimal for the dual problem (49); that is,

$$f^\star \in \underset{f \in \mathcal{F}}{\text{argmin}} \; \underset{\lambda \in \mathcal{B}(\mathcal{E}_{\text{all}})}{\max} \; R(f) + \int_{\mathcal{E}_{\text{all}}} [\mathcal{L}^e(\varphi(\theta, \cdot)) - \gamma] \, d\lambda(e) \tag{50}$$

and

$$\lambda^\star \in \underset{\lambda \in \mathcal{B}(\mathcal{E}_{\text{all}})}{\text{argmax}} \; \underset{f \in \mathcal{F}}{\min} \; R(f) + \int_{\mathcal{E}_{\text{all}}} [\mathcal{L}^e(\varphi(\theta, \cdot)) - \gamma] \, d\lambda(e) \tag{51}$$

At this early stage, it will be useful to state the following saddle-point relation, which is a direct result of strong duality:

$$\Lambda(f^\star, \lambda') \leq \Lambda(f^\star, \lambda^\star) \leq \Lambda(f', \lambda^\star) \tag{52}$$

which holds for all $f' \in \mathcal{F}$ and for all $\lambda' \in \mathcal{B}(\mathcal{E}_{\text{all}})$. Now consider that by the definition of the optimization problem in (4), we have that

$$D^{\star}_{\epsilon}(\gamma) = \max_{\lambda \in \mathcal{B}(\mathcal{E}_{\text{all}})} \min_{\theta \in \mathcal{H}} \Lambda(\theta, \lambda) \geq \min_{\theta \in \mathcal{H}} \Lambda(\theta, \lambda') \quad \forall \lambda' \in \mathcal{B}(\mathcal{E}_{\text{all}}). \tag{53}$$

Therefore, by choosing $\lambda' = \lambda^{\star}$ in the above expression, and since $\mathcal{A}_{\epsilon} = \{\varphi(\theta, \cdot) : \theta \in \mathcal{H}\} \subseteq \mathcal{F}$ by the definition of an $\epsilon$-parametric approximation, we have that

$$D^{\star}_{\epsilon}(\gamma) \geq \min_{\theta \in \mathcal{H}} \Lambda(\theta, \lambda^{\star}) \geq \min_{f \in \mathcal{F}} \Lambda(f, \lambda^{\star}) = P^{\star}(\gamma). \tag{54}$$

This concludes the proof of the lower bound: $P^{\star}(\gamma) \leq D^{\star}_{\epsilon}(\gamma)$.

**Step 2.** Next, we show that $D^{\star}_{\epsilon}(\gamma)$ is upper bounded by the optimal value of a perturbed version of the empirical dual problem. To begin, we add and subtract $\min_{f \in \mathcal{F}} \Lambda(f, \lambda)$ from the parameterized dual problem in (4).

$$D^{\star}_{\epsilon}(\gamma) = \max_{\lambda \in \mathcal{B}(\mathcal{E}_{\text{all}})} \min_{\theta \in \mathcal{H}} \left[ \Lambda(\theta, \lambda) + \min_{f \in \mathcal{F}} \Lambda(f, \lambda) - \min_{f \in \mathcal{F}} \Lambda(f, \lambda) \right] \tag{55}$$

$$= \max_{\lambda \in \mathcal{B}(\mathcal{E}_{\text{all}})} \min_{\substack{\theta \in \mathcal{H} \\ f \in \mathcal{F}}} \Lambda(f, \lambda) + \left[ R(\varphi(\theta, \cdot)) - R(f) \right] + \int_{\mathcal{E}_{\text{all}}} \left[ \mathcal{L}^{e}(\varphi(\theta, \cdot)) - \mathcal{L}^{e}(f) \right] d\lambda(e) \tag{56}$$

Now let $\mu(e)$ denote any probability measure with support over $\mathcal{E}_{\text{all}}$. Consider the latter two terms in the above problem, and observe that we can write

$$\left[ R(\varphi(\theta, \cdot)) - R(f) \right] + \int_{\mathcal{E}_{\text{all}}} \left[ \mathcal{L}^{e}(\varphi(\theta, \cdot)) - \mathcal{L}^{e}(f) \right] d\lambda(e) \tag{57}$$

$$= \int_{\mathcal{E}_{\text{all}}} \left[ R(\varphi(\theta, \cdot)) - R(f) \right] \mu(e) de + \int_{\mathcal{E}_{\text{all}}} \left[ \mathcal{L}^{e}(\varphi(\theta, \cdot)) - \mathcal{L}^{e}(f) \right] \lambda(e) de \tag{58}$$

$$= \int_{\mathcal{E}_{\text{all}}} \begin{bmatrix} R(\varphi(\theta, \cdot)) - R(f) \\ \mathcal{L}^{e}(\varphi(\theta, \cdot)) - \mathcal{L}^{e}(f) \end{bmatrix}^{\top} \begin{bmatrix} \mu(e) \\ \lambda(e) \end{bmatrix} de \tag{59}$$

$$\overset{(*)}{\leq} \int_{\mathcal{E}_{\text{all}}} \left\| \begin{bmatrix} \mu(e) \\ \lambda(e) \end{bmatrix} \right\|_{1} \cdot \left\| \begin{bmatrix} R(\varphi(\theta, \cdot)) - R(f) \\ \mathcal{L}^{e}(\varphi(\theta, \cdot)) - \mathcal{L}^{e}(f) \end{bmatrix} \right\|_{\infty} de \tag{60}$$

$$= \int_{\mathcal{E}_{\text{all}}} \left( \mu(e) + \lambda(e) \right) \cdot \max \left\{ R(\varphi(\theta, \cdot)) - R(f), \mathcal{L}^{e}(\varphi(\theta, \cdot)) - \mathcal{L}^{e}(f) \right\} de \tag{61}$$

$$\overset{(**)}{\leq} \left\| \mu + \lambda \right\|_{L^{1}} \cdot \left\| \max \left\{ R(\varphi(\theta, \cdot)) - R(f), \mathcal{L}^{e}(\varphi(\theta, \cdot)) - \mathcal{L}^{e}(f) \right\} \right\|_{L^{\infty}} \tag{62}$$

$$\overset{(\square)}{\leq} \left( 1 + \left\| \lambda \right\|_{L^{1}} \right) \cdot \left\| \max \left\{ R(\varphi(\theta, \cdot)) - R(f), \mathcal{L}^{e}(\varphi(\theta, \cdot)) - \mathcal{L}^{e}(f) \right\} \right\|_{L^{\infty}}. \tag{63}$$

where $(*)$ and $(**)$ follows from separate applications of Hölder's inequality [151], and $(\square)$ follows from an application of Minkowski's inequality and from the fact that $\mu$ is a (normalized) probability distribution. Let us now consider the second term in the above product:

$$\left\| \max \left\{ R(\varphi(\theta, \cdot)) - R(f), \mathcal{L}^{e}(\varphi(\theta, \cdot)) - \mathcal{L}^{e}(f) \right\} \right\|_{L^{\infty}} \tag{64}$$

$$= \left\| \max \{ \mathbb{E}[\ell(\varphi(\theta, X), Y) - \ell(f(X), Y)], \mathbb{E}[d(\varphi(\theta, X), \varphi(\theta, G(X, e))) - d(f(X), f(G(X, e)))] \} \right\|_{L^{\infty}} \tag{65}$$

$$\overset{(\circ)}{\leq} \left\| \mathbb{E} \left[ \max \{ |\ell(\varphi(\theta, X), Y) - \ell(f(X), Y)|, |d(\varphi(\theta, X), \varphi(\theta, G(X, e))) - d(f(X), f(G(X, e)))| \} \right] \right\|_{L^{\infty}} \tag{66}$$

$$\overset{(\triangle)}{\leq} \mathbb{E} \left\| \max \{ |\ell(\varphi(\theta, X), Y) - \ell(f(X), Y)|, |d(\varphi(\theta, X), \varphi(\theta, G(X, e))) - d(f(X), f(G(X, e)))| \} \right\|_{L^{\infty}} \tag{67}$$

$$\leq \mathbb{E} \left[ \max \{ L_{\ell} \left\| \varphi(\theta, X) - f(X) \right\|_{\infty}, L_{d} \left\| \varphi(\theta, X) - f(X) \right\|_{\infty} \} \right] \tag{68}$$

$$= \max \{ L_{\ell}, L_{d} \} \cdot \mathbb{E} \left\| \varphi(\theta, X) - f(X) \right\|_{\infty}. \tag{69}$$

where $(\circ)$ and $(\triangle)$ both follow from Jensen's inequality, and the final inequality follows from our Lipschitzness assumptions on $\ell$ and $d$. For simplicity, let $c = \max \{ L_{\ell}, L_{d} \}$. Now returning to (56),

we can combine (63) and (69) to obtain

$$D_\epsilon^\star(\gamma) \le \max_{\lambda \in \mathcal{B}(\mathcal{E}_{\mathrm{all}})} \min_{\substack{\theta \in \mathcal{H} \\ f \in \mathcal{F}}} \Lambda(f, \lambda) + c(1 + \|\lambda\|_{L^1}) \cdot \mathbb{E}\|\varphi(\theta, X) - f(X)\|_\infty \tag{70}$$

$$= \max_{\lambda \in \mathcal{B}(\mathcal{E}_{\mathrm{all}})} \min_{f \in \mathcal{F}} \Lambda(f, \lambda) + c(1 + \|\lambda\|_{L^1}) \cdot \min_{\theta \in \mathcal{H}} \mathbb{E}\|\varphi(\theta, X) - f(X)\|_\infty \tag{71}$$

$$\le \max_{\lambda \in \mathcal{B}(\mathcal{E}_{\mathrm{all}})} \min_{f \in \mathcal{F}} \Lambda(f, \lambda) + c\epsilon(1 + \|\lambda\|_{L^1}). \tag{72}$$

Now let $D_{\mathrm{pert}}^\star(\gamma)$ denote the optimal value of the above problem; that is,

$$D_{\mathrm{pert}}^\star(\gamma) \triangleq \max_{\lambda \in \mathcal{B}(\mathcal{E}_{\mathrm{all}})} \min_{f \in \mathcal{F}} \Lambda(f, \lambda) + c\epsilon(1 + \|\lambda\|_{L^1}) \tag{73}$$

$$= \max_{\lambda \in \mathcal{B}(\mathcal{E}_{\mathrm{all}})} \min_{f \in \mathcal{F}} R(f) + ce + \int_{\mathcal{E}_{\mathrm{all}}} [\mathcal{L}^e(f) - \gamma + c\epsilon]\, \mathrm{d}\lambda(e) \tag{74}$$

**Step 3.** In the final step, we prove the theorem. We begin with the perhaps unintuitive fact that the perturbed problem defined above is the dual problem to a perturbed version of the optimization problem in (3). More specifically, the perturbed problem in (74) is the dual of

$$P_{\mathrm{pert}}^\star(\gamma) \triangleq \underset{f \in \mathcal{F}}{\mathrm{minimize}} \quad R(f) + c\epsilon \tag{75}$$

$$\text{subject to} \quad \mathcal{L}^e(f) \le \gamma - c\epsilon \quad \forall e \in \mathcal{E}_{\mathrm{all}}. \tag{76}$$

Note that as this primal perturbed optimization problem is convex since (3) is convex, and by assumption strong duality also holds for this perturbed problem. Let $(f_{\mathrm{pert}}^\star, \lambda_{\mathrm{pert}}^\star)$ be primal-dual optimal for the perturbed problems we have defined above. The following saddle-point relation is evident from the fact that strong duality holds:

$$\Lambda(f_{\mathrm{pert}}^\star, \lambda') + c\epsilon\,(1 + \|\lambda'\|_{L^1}) \le D_{\mathrm{pert}}^\star(\gamma) = P_{\mathrm{pert}}^\star(\gamma) \le \Lambda(f', \lambda_{\mathrm{pert}}^\star) + c\epsilon\left(1 + \|\lambda_{\mathrm{pert}}^\star\|_{L^1}\right) \tag{77}$$

where the inequalities hold for all $f' \in \mathcal{F}$ and for all $\lambda' \in \mathcal{B}(\mathcal{E}_{\mathrm{all}})$. Using this result for the choice of $f' = f^\star$, where we recall that $f^\star$ is defined in (50) as the primal optimal solution to (3), it follows from (72) that

$$D_\epsilon^\star(\gamma) \le D_{\mathrm{pert}}^\star(\gamma) \le \Lambda(f^\star, \lambda_{\mathrm{pert}}^\star) + c\epsilon\left(1 + \|\lambda_{\mathrm{pert}}^\star\|_{L^1}\right) \tag{78}$$

Now, recalling the original saddle-point relation in (72), it holds that $\Lambda(f^\star, \lambda_{\mathrm{pert}}^\star) \le \Lambda(f^\star, \lambda^\star)$. Using this fact along with (78) yields the following result:

$$D_\epsilon^\star(\gamma) \le \Lambda(f^\star, \lambda^\star) + c\epsilon\left(1 + \|\lambda_{\mathrm{pert}}^\star\|_{L^1}\right) = P^\star(\gamma) + c\epsilon\left(1 + \|\lambda_{\mathrm{pert}}^\star\|_{L^1}\right) \tag{79}$$

This completes the proof. $\qquad\square$

### C.4 Characterizing the empirical gap (used in Theorem 5.3)

**Proposition C.2** (Empirical gap). *Assume $\ell$ and $d$ are non-negative and bounded in $[-B, B]$ and let $d_{\mathrm{VC}}$ denote the VC-dimension of the hypothesis class $\mathcal{A}_\epsilon$. Then it holds with probability $1 - \delta$ over the $N$ samples from each domain that*

$$|D_\epsilon^\star(\gamma) - D_{\epsilon, N, \mathcal{E}_{\mathrm{train}}}^\star(\gamma)| \le 2B\sqrt{\frac{1}{N}\left[1 + \log\left(\frac{4(2N)^{d_{\mathrm{vc}}}}{\delta}\right)\right]} \tag{80}$$

*Proof.* In this proof, we use a similar approach as in [142, Prop. 2] to derive the generalization bound. Notably, we extend the ideas given in this proof to accommodate two problems with different constraints, wherein the constraints of one problem are a strict subset of the other problem.

To begin, let $(\theta_\epsilon^\star, \lambda_\epsilon^\star)$ and $(\theta_{\epsilon, N, \mathcal{E}_{\mathrm{train}}}^\star, \lambda_{\epsilon, N, \mathcal{E}_{\mathrm{train}}}^\star)$ be primal-dual optimal pairs for (4) and (6) that achieve $D_\epsilon^\star(\gamma)$ and $D_{\epsilon, N, \mathcal{E}_{\mathrm{train}}}^\star(\gamma)$ respectively; that is,

$$(\theta_\epsilon^\star, \lambda_\epsilon^\star) \in \underset{\lambda \in \mathcal{P}(\mathcal{E}_{\mathrm{all}})}{\mathrm{argmax}} \min_{\theta \in \mathcal{H}} R(\varphi(\theta, \cdot)) + \int_{\mathcal{E}_{\mathrm{all}}} [\mathcal{L}^e(\varphi(\theta, \cdot)) - \gamma]\, \mathrm{d}\lambda(e). \tag{81}$$

and

$$(\theta^\star_{\epsilon,N,\mathcal{E}_{\text{train}}}, \lambda^\star_{\epsilon,N,\mathcal{E}_{\text{train}}}) \in \underset{\lambda(e) \geq 0,\, e \in \mathcal{E}_{\text{train}}}{\operatorname{argmax}} \min_{\theta \in \mathcal{H}} \hat{R}(\varphi(\theta, \cdot)) + \frac{1}{|\mathcal{E}_{\text{train}}|} \sum_{e \in \mathcal{E}_{\text{train}}} \left[ \hat{\mathcal{L}}^e(\varphi(\theta, \cdot)) - \gamma \right] \lambda(e) \quad (82)$$

are satisfied. Due to the optimality of these primal-dual pairs, both primal-dual pairs satisfy the KKT conditions [148]. In particular, the complementary slackness condition implies that

$$\int_{\mathcal{E}_{\text{all}}} \left[ \mathcal{L}^e(\varphi(\theta^\star_\epsilon, \cdot)) - \gamma \right] \mathrm{d}\lambda^\star_\epsilon(e) = 0 \quad (83)$$

and that

$$\frac{1}{|\mathcal{E}_{\text{train}}|} \sum_{e \in \mathcal{E}_{\text{train}}} \left[ \hat{\mathcal{L}}^e(\varphi(\theta^\star_{\epsilon,N,\mathcal{E}_{\text{train}}}, \cdot)) - \gamma \right] \lambda^\star_{\epsilon,N,\mathcal{E}_{\text{train}}}(e) = 0. \quad (84)$$

Thus, as (83) indicates that the second term in the objective of (81) is zero, we can recharacterize the optimal value $D^\star_\epsilon(\gamma)$ via

$$D^\star_\epsilon(\gamma) = R(\varphi(\theta^\star_\epsilon, \cdot)) = \mathbb{E}_{\mathbb{P}(X,Y)}\, \ell(\varphi(\theta^\star_\epsilon, X), Y) \quad (85)$$

and similarly from (84), can recharacterize the optimal value $D^\star_{\epsilon,N,\mathcal{E}_{\text{train}}}(\gamma)$ as

$$D^\star_{\epsilon,N,\mathcal{E}_{\text{train}}}(\gamma) = \hat{R}(\varphi(\theta^\star_{\epsilon,N,\mathcal{E}_{\text{train}}}, \cdot)) = \frac{1}{N} \sum_{i=1}^{N} \ell(\varphi(\theta^\star_{\epsilon,N,\mathcal{E}_{\text{train}}}, x_i), y_i). \quad (86)$$

Ultimately, our goal is to bound the gap between $|D^\star_\epsilon(\gamma) - D^\star_{\epsilon,N,\mathcal{E}_{\text{train}}}(\gamma)|$. Combining (85) and (86), we see that this gap can be characterized in the following way

$$|D^\star_\epsilon(\gamma) - D^\star_{\epsilon,N,\mathcal{E}_{\text{train}}}(\gamma)| = |R(\varphi(\theta^\star_\epsilon, \cdot)) - \hat{R}(\varphi(\theta^\star_{\epsilon,N,\mathcal{E}_{\text{train}}}, \cdot))|. \quad (87)$$

Now due to the optimality of the primal-optimal variables $\theta^\star_\epsilon$ and $\theta^\star_{\epsilon,N,\mathcal{E}_{\text{train}}}$, observe that

$$R(\varphi(\theta^\star_\epsilon, \cdot)) - \hat{R}(\varphi(\theta^\star_\epsilon, \cdot)) \quad (88)$$
$$\leq R(\varphi(\theta^\star_\epsilon, \cdot)) - \hat{R}(\varphi(\theta^\star_{\epsilon,N,\mathcal{E}_{\text{train}}}, \cdot)) \quad (89)$$
$$\leq R(\varphi(\theta^\star_{\epsilon,N,\mathcal{E}_{\text{train}}}, \cdot)) - \hat{R}(\varphi(\theta^\star_{\epsilon,N,\mathcal{E}_{\text{train}}}, \cdot)) \quad (90)$$

which, when combined with (87), implies that

$$|D^\star_\epsilon(\gamma) - D^\star_{\epsilon,N,\mathcal{E}_{\text{train}}}(\gamma)| \quad (91)$$
$$\leq \max \left\{ \left| R(\varphi(\theta^\star_\epsilon, \cdot)) - \hat{R}(\varphi(\theta^\star_\epsilon, \cdot)) \right|, \left| R(\varphi(\theta^\star_{\epsilon,N,\mathcal{E}_{\text{train}}}, \cdot)) - \hat{R}(\varphi(\theta^\star_{\epsilon,N,\mathcal{E}_{\text{train}}}, \cdot)) \right| \right\}. \quad (92)$$

To wrap up the proof, we simply leverage the classical VC-dimension bounds for both of the terms in (92). That is, following [43], it holds for all $\theta$ that with probability $1 - \delta$,

$$|R(\varphi(\theta, \cdot)) - \hat{R}(\varphi(\theta), \cdot)| \leq 2B \sqrt{\frac{1}{N} \left[ 1 + \log \left( \frac{4(2N)^{d_{\text{vc}}}}{\delta} \right) \right]}. \quad (93)$$

As the bound in (93) holds $\forall \theta \in \mathcal{H}$, in particular it holds for $\theta^\star_\epsilon$ and $\theta^\star_{\epsilon,N,\mathcal{E}_{\text{train}}}$. This directly implies the bound in (80). $\qquad \square$

## C.5   Proof of Theorem 5.3

The **Theorem 5.3.** Let $\epsilon > 0$ be given, and let $\varphi$ be an $\epsilon$-parameterization of $\mathcal{F}$. Let Assumption C.1 hold, and further assume that $\ell$ and $d$ are $[0, B]$-bounded and that $d(\mathbb{P}, \mathbb{T}) = 0$ if and only if $\mathbb{P} = \mathbb{T}$ almost surely, and that $P^\star(\gamma)$ is $L$-Lipschitz. Then assuming that $\mathcal{A}_\epsilon$ has finite VC-dimension, it holds with probability $1 - \delta$ over the $N$ samples from $\mathbb{P}$ that

$$|P^\star - D^\star_{\epsilon,N,\mathcal{E}_{\text{train}}}(\gamma)| \leq L\gamma + (L_\ell + 2L_d)\epsilon + \mathcal{O}\left( \sqrt{\log(N)/N} \right) \quad (94)$$

*Proof.* The proof of this theorem is a simple consequence of the triangle inequality. Indeed, by combining Remark B.2, Proposition 5.2, and Proposition C.2, we find that

$$|P^\star - D^\star_{\epsilon,N,\mathcal{E}_{\text{train}}}(\gamma)| \tag{95}$$

$$= |P^\star + P^\star(\gamma) - P^\star(\gamma) + D^\star_\epsilon(\gamma) - D^\star_\epsilon(\gamma) - D^\star_{\epsilon,N,\mathcal{E}_{\text{train}}}(\gamma)| \tag{96}$$

$$\leq |P^\star - P^\star(\gamma)| + |P^\star(\gamma) - D^\star_\epsilon(\gamma)| + |D^\star(\gamma) - D^\star_{\epsilon,N,\mathcal{E}_{\text{train}}}(\gamma)| \tag{97}$$

$$\leq L\gamma + \epsilon k \left(1 + \left\|\lambda^\star_{\text{pert}}\right\|_{L^1}\right) + 2B\sqrt{\frac{1}{N}\left[1 + \log\left(\frac{4(2N)^{d_{\text{VC}}}}{\delta}\right)\right]}. \tag{98}$$

This completes the proof. □

## C.6 Proof of Theorem 6.1

**Theorem 6.1.** Assume that $\ell$ and $d$ are $[0, B]$-bounded, convex, and $M$-Lipschitz continuous (i.e. $M = \max\{L_\ell, L_d\}$. Further, assume that $\mathcal{H}$ has finite VC-dimension $d_{\text{VC}}$ and that for each $\theta_1, \theta_2 \in \mathcal{H}$ and for each $\beta \in [0, 1]$, there exists a parameter $\theta \in \mathcal{H}$ and a constant $\nu > 0$ such that

$$\mathbb{E}_{\mathbb{P}(X,Y)} |\beta\varphi(\theta_1, X) + (1 - \beta)\varphi(\theta_2, X) - \varphi(\theta, X)| \leq \nu. \tag{99}$$

Finally, assume that there exists a parameter $\theta \in \mathcal{H}$ such that $\varphi(\theta, \cdot)$ is strictly feasible for (3), i.e. that

$$\mathcal{L}^e(\varphi(\theta, \cdot)) \leq \gamma - M\nu \quad \forall e \in \mathcal{E}_{\text{all}} \tag{100}$$

where $\nu$ is the constant from (99). Then it follows that the primal-dual pair $(\theta^{(T)}, \lambda^{(T)})$ obtained after running the alternating primal-dual iteration in (8) and (9) for $T$ steps with step size $\eta$, where

$$T \triangleq \left\lceil \frac{\|\lambda^\star\|}{2\eta M\nu} \right\rceil + 1 \quad \text{and} \quad \eta \leq \frac{2M\nu}{|\mathcal{E}_{\text{train}}|B^2} \tag{101}$$

satisfies

$$|P^\star - \hat{\Lambda}(\theta^{(T)}, \mu^{(T)})| \leq \rho + M\nu + L\gamma + \mathcal{O}(\sqrt{\log(N)/N}) \tag{102}$$

where $\|\lambda^\star\|$ is the optimal dual variable for (4).

*Proof.* Observe that by the triangle inequality, we have

$$|P^\star - \hat{\Lambda}(\theta^{(T)}, \mu^{(T)})| = |P^\star - P^\star(\gamma) + P^\star(\gamma) - \hat{\Lambda}(\theta^{(T)}, \mu^{(T)})| \tag{103}$$

$$\leq |P^\star - P^\star(\gamma)| + |P^\star(\gamma) - \hat{\Lambda}(\theta^{(T)}, \mu^{(T)})| \tag{104}$$

$$\leq L\gamma + |P^\star(\gamma) - \hat{\Lambda}(\theta^{(T)}, \mu^{(T)})| \tag{105}$$

where the last step follows from Remark B.2. Then, from [152, Theorem 2], it directly follows that

$$|P^\star(\gamma) - \hat{\Lambda}(\theta^{(T)}, \mu^{(T)})| \leq \rho + M\nu + \mathcal{O}\sqrt{\log(N)/N}. \tag{106}$$

Combining this with (105) completes the proof. □

---

**Algorithm 2** ERM with model-based data augmentation (MBDA)

---

1: **Hyperparameters:** Step size $\eta > 0$
2: **repeat**
3:     **for** minibatch $\{(x_j, y_j)\}_{j=1}^m$ in training dataset **do**
4:         $\tilde{x}_j \leftarrow \textsc{GenerateImage}(x_j) \; \forall j \in [m]$             ▷ Generate model-based images
5:         $\text{loss}(\theta) \leftarrow (1/m) \sum_{j=1}^m [\ell(x_j, y_j; \varphi(\theta, \cdot)) + \ell(\tilde{x}_j, y_j; \varphi(\theta, \cdot))]$
6:         $\theta \leftarrow \theta - \eta \nabla_\theta \text{loss}(\theta)$
7:     **end for**
8: **until** convergence

---

---

**Algorithm 3** MBDG with data augmentation (MBDG-DA)

---

1: **Hyperparameters:** Primal step size $\eta_p > 0$, dual step size $\eta_d \geq 0$, margin $\gamma > 0$
2: **repeat**
3:     **for** minibatch $\{(x_j, y_j)\}_{j=1}^m$ in training dataset **do**
4:         $\tilde{x}_j \leftarrow \textsc{GenerateImage}(x_j) \; \forall j \in [m]$        ▷ Generate images for constraints
5:         $\overline{x}_j \leftarrow \textsc{GenerateImage}(x_j) \; \forall j \in [m]$        ▷ Generate images for objective
6:         $\text{loss}(\theta) \leftarrow (1/m) \sum_{j=1}^m [\ell(x_j, y_j; \varphi(\theta, \cdot)) + \ell(\overline{x}_j, y_j; \varphi(\theta, \cdot)) + \ell(\tilde{x}_j, y_j; \varphi(\theta, \cdot))]$
7:         $\text{distReg}(\theta) \leftarrow (1/m) \sum_{j=1}^m d(\varphi(\theta, x_j), \varphi(\theta, \tilde{x}_j))$
8:         $\theta \leftarrow \theta - \eta_p \nabla_\theta [\, \text{loss}(\theta) + \lambda \cdot \text{distReg}(\theta) \,]$
9:         $\lambda \leftarrow [\lambda + \eta_d (\text{distReg}(\theta) - \gamma)]_+$
10:     **end for**
11: **until** convergence

---

# D   Algorithmic variants for MBDG

In Section 7, we considered several algorithmic variants of MBDG. Each variant offers a natural point of comparison to the MBDG algorithm, and for completeness, in this section we fully characterize these variants.

## D.1   Data augmentation

In Section 7, we did an ablation study concerning various data-augmentation alternatives to MBDG. In particular, in the experiments performed on `ColoredMNIST`, we compared results obtained with MBDG to two algorithms we called MBDA and MBDG-DA. For clarity, in what follows we describe each of them in more detail.

**MBDA.** In the MDBA variant, we train using ERM with data augmentation through the learned domain transformation model $G(x, e)$. This procedure is summarized in Algorithm 2. Notice that in this algorithm, we do not consider the constraints engendered by the assumption of $G$-invariance. Rather, we simply seek to use follow the recent empirical evidence that suggests that ERM with proper tuning and data augmentation yields state-of-the-art performance in domain generalization [46]. Note that in Table 1, the MBDA algorithm improves significantly over the baselines, but that it lags more than 20 percentage points behind results obtained using MBDG. This highlights the utility of enforcing constraints rather than performing data augmentation on the training objective.

**MBDG-DA.** In the MBDG-DA variant, we follow a similar procedure to the MBDG algorithm. The only modification is that we perform data augmentation through the learned model $G(x, e)$ on the training objective in addition to enforcing the $G$-invariance constraints. This procedure is summarized in Algorithm 3. As shown in Table 1, this procedure performs rather well on `ColoredMNIST`, beating all baselines by nearly 20 percentage points. However, this algorithm still does not reach the performance level of MBDG when the -90% domain is taken to be the test domain.

**Algorithm 4** Regularized MBDG (MBDG-Reg)

---

1: **Hyperparameters:** Step size $\eta > 0$, weight $w > 0$
2: **repeat**
3:     **for** minibatch $\{(x_j, y_j)\}_{j=1}^m$ in training dataset **do**
4:         $\tilde{x}_j \leftarrow \textsc{GenerateImage}(x_j) \; \forall j \in [m]$                 $\triangleright$ Generate model-based images
5:         $\text{loss}(\theta) \leftarrow (1/m) \sum_{j=1}^m [\ell(x_j, y_j; \varphi(\theta, \cdot)) + \ell(\tilde{x}_j, y_j; \varphi(\theta, \cdot))]$
6:         $\text{distReg}(\theta) \leftarrow (1/m) \sum_{j=1}^m d(\varphi(\theta, x_j), \varphi(\theta, \tilde{x}_j))$
7:         $\theta \leftarrow \theta - \eta \nabla_\theta [\text{loss}(\theta) + w \cdot \text{distReg}(\theta)]$
8:     **end for**
9: **until** convergence

---

### D.2 Regularization

In Section 7, we also compared the performance of MBDG to a regularized version of MBDG. In this regularized version, we sought to solve (32) using the algorithm described in Algorithm 4. In particular, in this algorithm we fix the weight $w > 0$ as a hyperparameter, and we perform SGD on the regularized loss function $\text{loss}(\theta) + w \cdot \text{distReg}(\theta)$. Note that while this method performs well in practice (see Table 1), it is generally not possible to provide generalization guarantees for the regularized version of the problem.

Table 4: **DomainBed hyperparameters for MBDG and its variants.** We record the additional hyperparameters and their selection criteria for MBDG and its variants. Each of these hyperparameters was selected via randomly in the ranges defined in the third column in the DomainBed package.

| Algorithm | Hyperparameter | Randomness | Default |
|---|---|---|---|
| MBDG | Dual step size $\eta_d$ | Unif$(0.001, 0.1)$ | 0.05 |
| | Constraint margin $\gamma$ | Unif$(0.0001, 0.01)$ | 0.025 |
| MBDG-DA | Dual step size $\eta_d$ | Unif$(0.001, 0.1)$ | 0.05 |
| | Constraint margin $\gamma$ | Unif$(0.0001, 0.01)$ | 0.025 |
| MBDG-Reg | Weight $w$ | Unif$(0.5, 10.0)$ | 1.0 |

# E    Additional experiments and experimental details

In this appendix, we record further experimental details beyond the results presented in Section 7. The experiments performed on `ColoredMNIST`, `PACS`, and `VLCS` were all performed using the DomainBed package[4]. All of the default hyperparameters (e.g. learning rate, weight decay, etc.) were left unchanged from the standard DomainBed implementation. In Table 4, we record the additional hyperparameters used for MBDG and its variants as well as the random criteria by which hyperparameters were generated. For each of these DomainBed datasets, model-selection was performed via hold-one-out cross-validation, and the baseline accuracies were taken from commit 7df6f06 of the DomainBed repository. The experiments on the `WILDS` datasets used the hyperparameters recorded by the authors of [20]; these hyperparameters are recorded in Sections E.1 and E.2. Throughout the experiments, we use the KL-divergence as the distance metric $d$.

## E.1   Camelyon17-WILDS

For the `Camelyon17-WILDS` dataset, we used the out-of-distribution validation set provided in the `Camelyon17-WILDS` dataset to tune the hyperparameters for each classifier. This validation set contains images from a hospital that is not represented in any of the training domains or the test domain. Following [20], we used the DenseNet-121 architecture [45] and the Adam optimizer [153] with a batch size of 200. We also used the same hyperparameter sweep as was described in Appendix B.4 of [20]. In particular, when training using our algorithm, we used the the following grid for the (primal) learning rate: $\eta_p \in \{0.01, 0.001, 0.0001\}$. Because we use the same hyperparameter sweep, architecture, and optimizer, we report the classification accuracies recorded in Table 9 of [20] to provide a fair comparison to past work. After selecting the hyperparameters based on the accuracy on the validation set, we trained classifiers using MBDG for 10 independent runs and reported the average accuracy and standard deviation across these trials in Table 2.

In Section 7, we performed an ablation study on `Camelyon17-WILDS` wherein the model $G$ was replaced by standard data augmentation transforms. For completeness, we describe each of the methods used in this plot below. For each method, invariance was enforced between a clean images drawn from the training domains and corresponding data that was varied according to a particular fixed transformation.

**CJ (Color Jitter).** The PIL color transformation[5]. See Figure 8 for samples.

**B+C (Brightness and contrast).** PIL `Brightness`[6] and `Contrast`[7] transformations. See Figure 9 for samples.

**RA (RandAugment).** We use the data augmentation technique RandAugment [145], which randomly samples random transformations to be applied at training time. In particular, the fol-

---

[4] `https://github.com/facebookresearch/DomainBed`
[5] `https://pillow.readthedocs.io/en/stable/reference/ImageEnhance.html#PIL.ImageEnhance.Color`
[6] `https://pillow.readthedocs.io/en/stable/reference/ImageEnhance.html#PIL.ImageEnhance.Brightness`
[7] `https://pillow.readthedocs.io/en/stable/reference/ImageEnhance.html#PIL.ImageEnhance.Contrast`

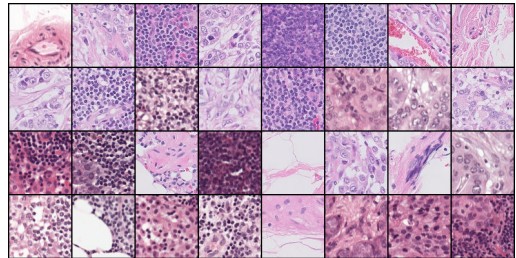 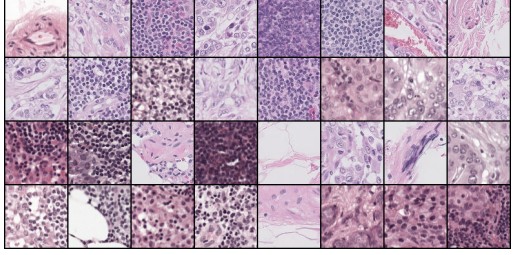

(a) Training images.  (b) Corresponding images after augmentations.

Figure 8: **Samples before and after CJ transformations.**

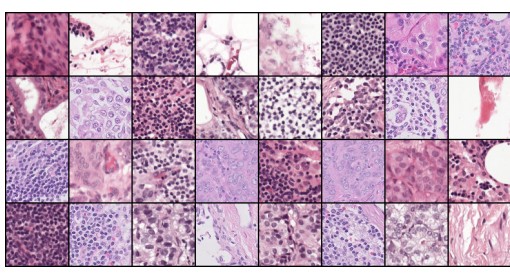 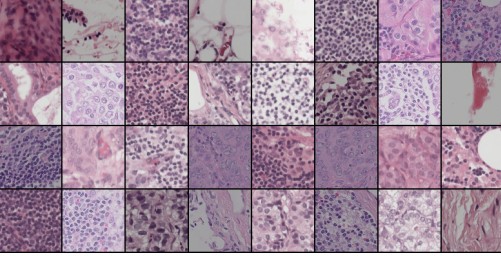

(a) Training images.  (b) Corresponding images after augmentations.

Figure 9: **Samples before and after B+C transformations.**

lowing transformations are randomly sampled: `AutoContrast`, `Equalize`, `Invert`, `Rotate`, `Posterize`, `Solarize`, `SolarizeAdd`, `Color`, `Constrast`, `Brightness`, `Sharpness`, `ShearX`, `ShearY`, `CutoutAbs`, `TranslateXabs`, and `TranslateYabs`. We used an open-source implementation of RandAugment for this experiment[8]. See Figure 10 for samples.

**RA-Geom (RandAugment with geometric transformations).** We use the RandAugment scheme with a subset of the transformations mentioned in the previous paragraph. In particular, we use the following geometric transformations: `Rotate`, `ShearX`, `ShearY`, `CutoutAbs`, `TranslateXabs`, and `TranslateYabs`. See Figure 11 for samples.

**RA-Color (RandAugment with color-based transformations).** We use the RandAugment scheme with a subset of transformations mentioned in the RandAugment paragraph. In particular, we use the following color-based transformations: `AutoContrast`, `Equalize`, `Invert`, `Posterize`, `Solarize`, `SolarizeAdd`, `Color`, `Constrast`, `Brightness`, `Sharpness`. See Figure 12 for samples.

**MUNIT.** We use an MUNIT model trained on the images from the training datasets; this is the procedure advocated for in the main text, i.e. in the GENERATEIMAGE(x) procedure. See Figure 13 for samples.

### E.2 FMoW-WILDS

As with the `Camelyon17-WILDS` dataset, to facilitate a fair comparison, we again use the out-of-distribution validation set provided in [20]. While the authors report the architecture, optimizer, and final hyperparameter choices used for the `FMoW-WILDS` dataset, they not report the grid used for hyperparameter search. For this reason, we rerun all baselines along with our algorithm over a grid of hyperparameters using the same architecture and optimizer as in [20]. In particular, we follow [20] by training a DenseNet-121 with the Adam optimizer with a batch size of 64. We selected the (primal) learning rate from $\eta_p \in \{0.05, 0.01, 0.005, 0.001\}$. We selected the trade-off parameter $\lambda_{\text{IRM}}$ for IRM from the grid $\lambda_{\text{IRM}} \in \{0.1, 0.5, 1.0, 10.0\}$. As before, the results in Table 2 list the average accuracy and standard deviation over ten independent runs attained by our algorithm as well as ERM, IRM, and ARM.

---

[8] https://github.com/ildoonet/pytorch-randaugment

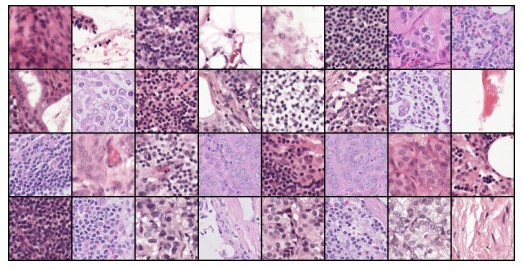

(a) Training images.

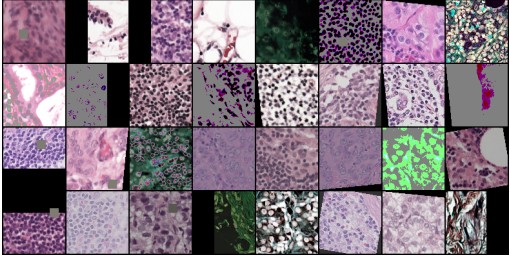

(b) Corresponding images after augmentations.

Figure 10: **Samples before and after RandAugment transformations.**

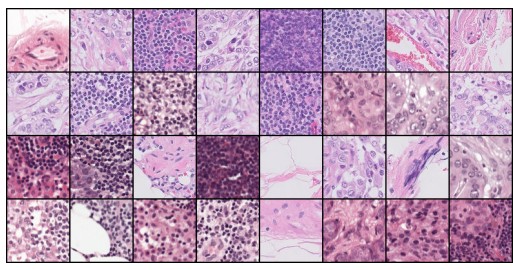

(a) Training images.

(b) Corresponding images after augmentations.

Figure 11: **Samples before and after RA-Geom transformations.**

### E.3 PACS

In Table 3, we provide a full set of results for the PACS dataset. Note that our result of 85.6% (averaged across the domains) is the best known result on PACS. In particular, this result is nearly two percentage points higher than any of the baselines, which represents a significant advancement in the state-of-the-art for this benchmark. In large part, this result is due to significant improvements on the "Sketch" (S) subset, wherein MBDG improves by nearly seven percentage points over all other baselines.

Table 5: **Full results for PACS.** In this table, we present results for all baselines on the PACS dataset.

| Algorithm | A | C | P | S | Avg |
|---|---|---|---|---|---|
| ERM | $83.2 \pm 1.3$ | $76.8 \pm 1.7$ | $\mathbf{97.2} \pm \mathbf{0.3}$ | $74.8 \pm 1.3$ | 83.0 |
| IRM | $81.7 \pm 2.4$ | $77.0 \pm 1.3$ | $96.3 \pm 0.2$ | $71.1 \pm 2.2$ | 81.5 |
| GroupDRO | $84.4 \pm 0.7$ | $77.3 \pm 0.8$ | $96.8 \pm 0.8$ | $75.6 \pm 1.4$ | 83.5 |
| Mixup | $85.2 \pm 1.9$ | $77.0 \pm 1.7$ | $96.8 \pm 0.8$ | $73.9 \pm 1.6$ | 83.2 |
| MLDG | $81.4 \pm 3.6$ | $77.9 \pm 2.3$ | $96.2 \pm 0.3$ | $76.1 \pm 2.1$ | 82.9 |
| CORAL | $80.5 \pm 2.8$ | $74.5 \pm 0.4$ | $96.8 \pm 0.3$ | $78.6 \pm 1.4$ | 82.6 |
| MMD | $84.9 \pm 1.7$ | $75.1 \pm 2.0$ | $96.1 \pm 0.9$ | $76.5 \pm 1.5$ | 83.2 |
| DANN | $84.3 \pm 2.8$ | $72.4 \pm 2.8$ | $96.5 \pm 0.8$ | $70.8 \pm 1.3$ | 81.0 |
| CDANN | $78.3 \pm 2.8$ | $73.8 \pm 1.6$ | $96.4 \pm 0.5$ | $66.8 \pm 5.5$ | 78.8 |
| MTL | $\mathbf{85.6} \pm 1.5$ | $78.9 \pm 0.6$ | $97.1 \pm 0.3$ | $73.1 \pm 2.7$ | 83.7 |
| SagNet | $81.1 \pm 1.9$ | $75.4 \pm 1.3$ | $95.7 \pm 0.9$ | $77.2 \pm 0.6$ | 82.3 |
| ARM | $85.9 \pm 0.3$ | $73.3 \pm 1.9$ | $95.6 \pm 0.4$ | $72.1 \pm 2.4$ | 81.7 |
| VREx | $81.6 \pm 4.0$ | $74.1 \pm 0.3$ | $96.9 \pm 0.4$ | $72.8 \pm 2.1$ | 81.3 |
| RSC | $83.7 \pm 1.7$ | $\mathbf{82.9} \pm \mathbf{1.1}$ | $95.6 \pm 0.7$ | $68.1 \pm 1.5$ | 82.6 |
| MBDG | $80.6 \pm 1.1$ | $79.3 \pm 0.2$ | $97.0 \pm 0.4$ | $\mathbf{85.2} \pm \mathbf{0.2}$ | $\mathbf{85.6}$ |

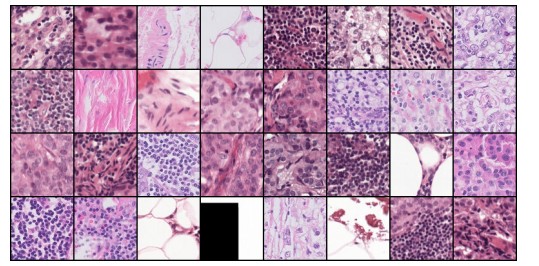 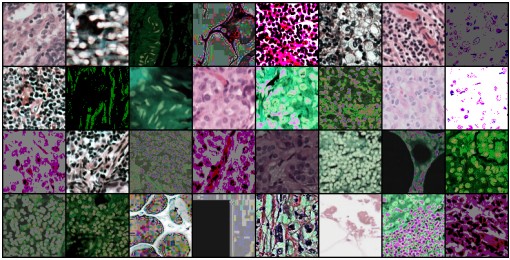

(a) Training images.  (b) Corresponding images after augmentations.

Figure 12: **Samples before and after RA-Color transformations.**

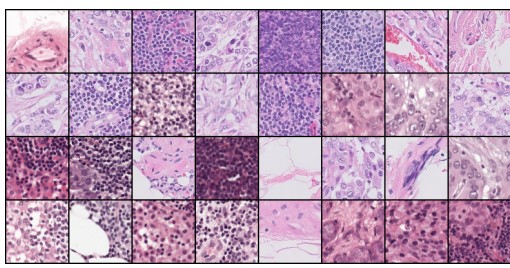 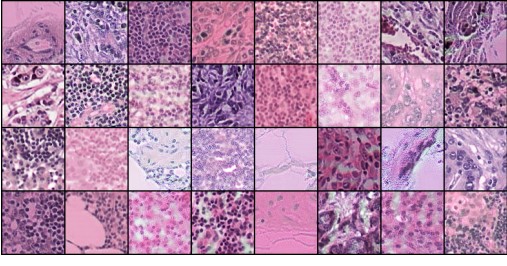

(a) Training images.  (b) Corresponding images after augmentations.

Figure 13: **Samples before and after (learned) MUNIT transformations.**

### E.4 VLCS

In Table 6, we provide a full set of results for the VLCS dataset. As shown in this Table, MBDG offers competitive performance to other state-of-the-art method. Indeed, MBDG achieves the best results on the "LabelMe" (L) subset by nearly two percentage points.

Table 6: **Full results for VLCS.** In this table, we present results for all baselines on the VLCS dataset.

| Algorithm | C | L | S | V | Avg |
|---|---|---|---|---|---|
| ERM | $98.0 \pm 0.4$ | $62.6 \pm 0.9$ | $70.8 \pm 1.9$ | $77.5 \pm 1.9$ | 77.2 |
| IRM | $\mathbf{98.6 \pm 0.3}$ | $66.0 \pm 1.1$ | $69.3 \pm 0.9$ | $71.5 \pm 1.9$ | 76.3 |
| GroupDRO | $98.1 \pm 0.3$ | $66.4 \pm 0.9$ | $71.0 \pm 0.3$ | $76.1 \pm 1.4$ | 77.9 |
| Mixup | $98.4 \pm 0.3$ | $63.4 \pm 0.7$ | $72.9 \pm 0.8$ | $76.1 \pm 1.2$ | 77.7 |
| MLDG | $98.5 \pm 0.3$ | $61.7 \pm 1.2$ | $\mathbf{73.6 \pm 1.8}$ | $75.0 \pm 0.8$ | 77.2 |
| CORAL | $96.9 \pm 0.9$ | $65.7 \pm 1.2$ | $73.3 \pm 0.7$ | $\mathbf{78.7 \pm 0.8}$ | **78.7** |
| MMD | $98.3 \pm 0.1$ | $65.6 \pm 0.7$ | $69.7 \pm 1.0$ | $75.7 \pm 0.9$ | 77.3 |
| DANN | $97.3 \pm 1.3$ | $63.7 \pm 1.3$ | $72.6 \pm 1.4$ | $74.2 \pm 1.7$ | 76.9 |
| CDANN | $97.6 \pm 0.6$ | $63.4 \pm 0.8$ | $70.5 \pm 1.4$ | $78.6 \pm 0.5$ | 77.5 |
| MTL | $97.6 \pm 0.6$ | $60.6 \pm 1.3$ | $71.0 \pm 1.2$ | $77.2 \pm 0.7$ | 76.6 |
| SagNet | $97.3 \pm 0.4$ | $61.6 \pm 0.8$ | $73.4 \pm 1.9$ | $77.6 \pm 0.4$ | 77.5 |
| ARM | $97.2 \pm 0.5$ | $62.7 \pm 1.5$ | $70.6 \pm 0.6$ | $75.8 \pm 0.9$ | 76.6 |
| VREx | $96.9 \pm 0.3$ | $64.8 \pm 2.0$ | $69.7 \pm 1.8$ | $75.5 \pm 1.7$ | 76.7 |
| RSC | $97.5 \pm 0.6$ | $63.1 \pm 1.2$ | $73.0 \pm 1.3$ | $76.2 \pm 0.5$ | 77.5 |
| MBDG | $98.3 \pm 1.2$ | $\mathbf{68.1 \pm 0.5}$ | $68.8 \pm 1.1$ | $76.3 \pm 1.3$ | 77.9 |

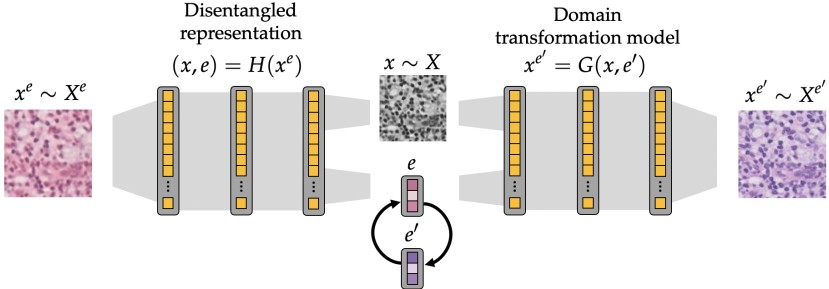

Figure 14: **Multi-modal image-to-image translation networks.** In this paper, we parameterize domain transformation models via multi-modal image-to-image translation networks, which can be trained to map images from one domain so that they resemble images from different domains.

## F    Learning domain transformation models from data

Regarding challenge (C4), critical to our approach is having access to the underlying domain transformation model $G(x, e)$. For the vast majority of settings, the underlying function $G(x, e)$ is not known a priori and cannot be represented by a simple expression. For example, obtaining a closed-form expression for a model that captures the variation in coloration, brightness, and contrast in the medical imaging dataset shown in Figure 1 would be challenging.

### F.1    Multimodal image-to-image translation networks

To address this challenge, we argue that a realistic *approximation* of the underlying domain transformation model can be learned from the instances drawn from the training datasets $\mathcal{D}^e$ for $e \in \mathcal{E}_{\text{train}}$. In this paper, to learn domain transformation models, we train multimodal image-to-image translation networks (MIITNs) on the instances drawn from the training domains. MIITNs are designed to transform samples from one dataset so that they resemble a diverse collection of images from another dataset. That is, the constraints used to train these models enforce that a diverse array of samples is outputted for each input image. This feature precludes the possibility of learning trivial maps between domains, such as the identity transformation.

As illustrated in Figure 14, these architectures generally consist of two components: a disentangled representation [154] and a generative model. The role of the disentangled representation is to recover a sample $x$ generated according to $X$ from a instance $x^e$ observed in a particular domain $e \in \mathcal{E}_{\text{all}}$. In other words, for a fixed instance $x^e = G(x, e)$, the disentangled representation is designed to disentangle $x$ from $e$ via $(x, e) = H(x^e)$. On the other hand, the role of the generative is to map each instance $x \sim X$ to a realization in a new environment $e'$. Thus, given $x$ and $e$ at the output of the disentangled representation, we generate an instance from a new domain by replacing the environmental code $e$ with a different environmental parameter $e' \in \mathcal{E}_{\text{all}}$ to produce the instance $x^{e'} = G(x, e')$. In this way, MIITNs are a natural framework for learning domain transformation models, as they facilitate 1) recovering samples from $X$ via the disentangled representation, and 2) generating instances from new domains in a multimodal fashion.

**Samples from learned domain transformation models.**    In each of the experiments in Section 7, we use the MUNIT architecture introduced in [102] to parameterize MIITNs. As shown in Table 7 and in Appendix G, models trained using the MUNIT architecture learn accurate and diverse transformations of the training data, which often generalize to generate images from new domains. Notice that in this table, while the generated samples still retain the characteristic features of the input image (e.g. in the top row, the cell patterns are the same across the generated samples), there is clear variation between the generated samples. Although these learned models cannot be expected to capture the full range of inter-domain generalization in the unseen test domains $\mathcal{E}_{\text{all}} \backslash \mathcal{E}_{\text{train}}$, in our experiments, we show that these learned models are sufficient to significantly advance the state-of-the-art on several domain generalization benchmarks.

Table 7: We show samples from domain transformation models trained on images from the training datasets $\mathcal{D}^e$ for $e \in \mathcal{E}_{\text{train}}$ using the MUNIT architecture for the `Camelyon17-WILDS`, `FMOW-WILDS`, and `PACS` datasets.

| Dataset | Original | Samples from learned domain transformation models $G(x, e)$ | | | |
|---|---|---|---|---|---|
| `ColoredMNIST` | | | | | |
| `Camelyon17-WILDS` | | | | | |
| `FMoW-WILDS` | | | | | |
| `PACS` | | | | | |

## G   Further discussion of domain transformation models

In some applications, domain transformation models in the spirit of Assumption 4.1 are known a priori. To illustrate this, consider the classic domain generalization task in which the domains correspond to different fixed rotations of the data [155, 57]. In this setting, the underlying generative model is given by

$$G(x, e) := R(e)x \quad \text{for } e \in [0, 2\pi) \tag{107}$$

where $R(e)$ is a one-dimensional rotation matrix parameterized by an angle $e$. In this way, each angle $e$ is identified with a different domain in $\mathcal{E}_{\text{all}}$. However, unlike in this simple example, for the vast majority of settings encountered in practice, the underlying domain transformation model is not known a priori and cannot be represented by concise mathematical expressions. For example, obtaining a closed-form expression for a generative model that captures the variation in coloration, brightness, and contrast in the `Camelyon17-WILDS` cancer cell dataset shown in Figure 1a would be very challenging.

In this appendix, we provide an extensive discussion concerning the means by which we used unlabeled data to learn domain transformation models using instances drawn from the training domains $\mathcal{E}_{\text{train}}$. In particular, we argue that it is not necessary to have access to the true underlying domain transformation model $G$ to achieve state-of-the-art results in domain generalization. We then give further details concerning how we used the MUNIT architecture to train domain transformation models for `ColoredMNIST`, `Camelyon17-WILDS`, `FMOW-WILDS`, `PACS`, and `VLCS`. Finally, we show further samples from these learned domain transformation models to demonstrate that high-quality samples can be obtained on this diverse array of datasets.

### G.1   Is it necessary to learn a perfect domain transformation model?

We emphasize that while our theoretical results rely on having access to the underlying domain transformation model, our algorithm and empirical results do not rely on having access to the true $G$. Indeed, although we did not have access to the true model in any of the experiments in Section 7, our empirical results show that we were able to achieve state-of-the-art results on several datasets.

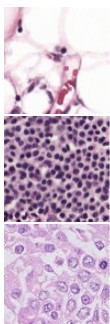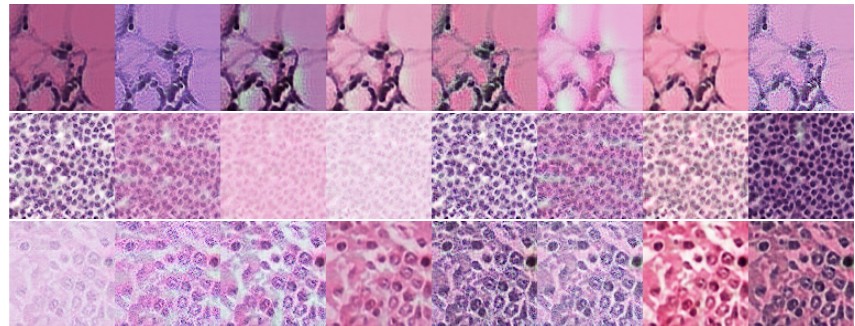

Figure 15: **Multimodal** `Camelyon17-WILDS` **samples.** Images from `Camelyon17-WILDS` (left) and images generated by sampling different style codes $e \sim \mathcal{N}(0, I)$ (right).

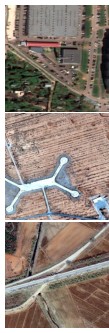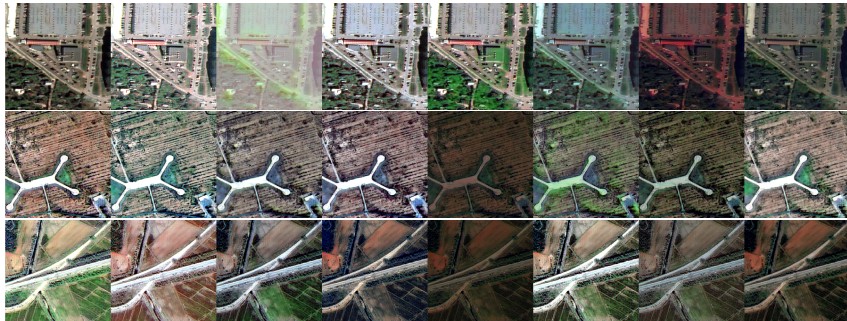

Figure 16: **Multimodal** `FMoW-WILDS` **samples.** Images from `FMoW-WILDS` (left) and images generated by sampling different style codes $e \sim \mathcal{N}(0, I)$ (right).

### G.2    Learning domain transformation models with MUNIT

In practice, to learn a domain transformation model, a number of methods from the deep generative modeling literature have been recently been proposed [102, 156, 157]. In particular, throughout the remainder of this paper we will use the MUNIT architecture introduced in [102] to parameterize learned domain transformation models. This architecture comprises two GANs and two autoencoding networks. In particular, the MUNIT architecture – along with many related works in the image-to-image translation literature – was designed to map images between two datasets $A$ and $B$. In this paper, rather than separating data we simply use $\mathcal{D}_X$ for both $A$ and $B$, meaning that we train MUNIT to map the training data back to itself. In this way, since $\mathcal{D}_X$ contains data from different domains $e \in \mathcal{E}_{\text{train}}$, the architecture is exposed to different environments during training, and thus seeks to map data between domains.

### G.3    On the utility of multi-modal image-to-image translation networks.

In this paper, we chose the MUNIT framework because it is designed to learn a multimodal transformation that maps an image $x$ to a family of images with different levels of variation. Unlike methods that seek deterministic mappings, e.g. CycleGAN and its variants [101], this method will learn to generate diverse images, which allows us to more effectively enforce invariance over a wider class of images. In Figures 15, 16, and 17, we plot samples generated by sampling different style codes $e \sim \mathcal{N}(0, I)$ for MUNIT. Note that while the results for `Camelyon17-WILDS` and `FMoW-WILDS` are sampled using the model $G(x, e)$, the samples from `PACS` are all sampled from *different* models.

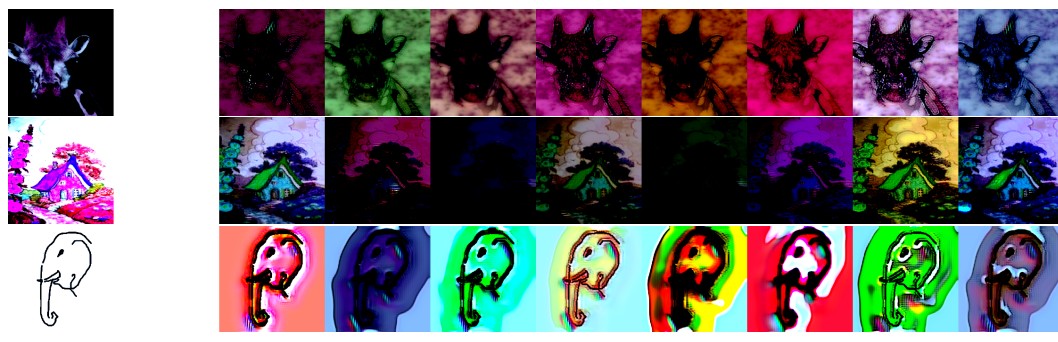

Figure 17: **Multimodal** `PACS` **samples.** Images from `PACS` (left) and images generated by sampling different style codes $e \sim \mathcal{N}(0, I)$ (right).