# OpenReview forum: "Model-Based Domain Generalization"
_NeurIPS.cc/2021/Conference — NeurIPS 2021 Poster_

### Official Review · Reviewer_VPwo · 2021-07-14

**Rating:** 6
**Confidence:** 3

**Summary:**

This paper proposes a framework for domain generalization which involves learning a transformation between domains and optimizing the worst case loss over these transformations. They formulate this as a min max optimization problem and eventually solve the dual of a relaxed version of the problem. Empirically, they find that their algorithm improves a lot for ColoredMNIST and Camelyon-WILDS especially, which are datasets where the domain transformations are somewhat simple (color shift). They get a modest gain on a satellite dataset FMoW-WILDS.

**Limitations And Societal Impact:**


While invariance and generalization to new domains could improve real world tasks like Camelyon, the authors mention that invariance is not always a desirable property, and models could be overly invariant / enforcing invariance can worsen extrapolation or at least change the inductive bias. They do not believe that there will be negative societal impacts relative to the body of work that already exists.



**Main Review:**

Significance

Domain generalization is an important problem, although it's unclear whether it is feasible to make meaningful progress. Implicitly, this paper shows that for particular domain generalization problems with simple transformation shifts, we can learn these transformations well and generalize OOD. Indeed, we shouldn't expect to be able to learn transformations between domains for general shifts, which can be very complex and hard to generalize across domains. The paper could state this more clearly, that the method needs to take advantage of some structure in the transformations (like color shift).

Originality

The overall setup is quite similar to [1], which learns generative perturbations of the data for adversarial learning. The difference is that the learning perturbations here use the different domains. [1] uses PGD adversarial training to optimize the min-max problem; this brings to question whether the primal-dual iterations are needed in the MBDG framework, and whether the problem in Prop 4.3 can be optimized directly through adversarial training.

Quality

The paper has some good ablations, for example it shows empirically that learned transformations on the training domains (using MUNIT) improve over fixed augmentations such as RandAugment for the MBDG framework. They give a mathematical derivation of the framework through the language of optimization, although it's unclear whether we can get theoretical insights from the general mathematical framework beyond suggesting an optimization algorithm (which perhaps could be replaced with adversarial training?).

- When testing the regularization version of MBDG, the paper shows that with $\lambda =1.0$ the performance isn't as good as MBDG with primal-dual iterations, but it's not clear whether $\lambda$ is varied. A proper comparison would have a sweep over $\lambda$, with some protocol for selecting $\lambda$ (some kind of validation accuracy, either ID or OOD).
- While the abstract states improvements "up to 30\%", the range of improvement is pretty wide (30\% on ColoredMNIST. 20\% on Camelyon17 but 1% on FMoW), and care should be taken when presenting these results faithfully.
- According to Figure 4, RandAugment with color is quite effective with MBDG for Camelyon. What is the comparison with ERM + RandAugment?
- The MBDG model improves on average over the PACS domains, but the improvement is mostly driven by improvement on sketch. This suggests that the MBDG model is best suited for particular shifts (yet to be characterized).
- For example, the G-invariance is the main special part of the theory, and it's clear from the relaxation formulation that when the G-invariance margin is small, then the gap is smaller between the relaxed problem and the original. The theory has some limitations, like that the hypothesis classes are of finite VC dimension, so it doesn't hold for deep models.
- G-invariance seems stronger than what is needed to solve the problem in Prop 4.3. In particular, it isn't necessary that $f(x) = f(G(x,e))$, but that for all $e, e' \in \varepsilon_{all}$, $f(G(x,e)) = f(G(x,e'))$. Maybe G-invariance works better in practice, but it's unclear from a formulation standpoint why this is formulated this way.

Clarity

The paper is clearly written, although the paper seems a little unfinished, with no conclusion. Secion 7.3 (PACS) lacks an adequate description of the task.


[1] Eric Wong and J. Zico Kolter. Learning Perturbation Sets for Robust Machine Learning, 2020.

**Time Spent Reviewing:**

3

---

> ### Author Response · Authors · 2021-08-10
> **Adversarial training as an alternative to primal-dual iteration + new ablation study with RandAugment + VC-dimension of NNs**
>
>
>
> **Response summary.**  Thanks for your positive review of our work!  We're glad that you liked our ablation studies and that you thought the paper was clearly written.  Based on your review, it seems that you have three main questions:
>
> 1. **Arbitrary shifts.**  Should we expect our method to be able to learn transformations between domains for general/arbitrary shifts?
>
> 2. **Adversarial training.** Would applying adversarial training (AT) on the minimax problem be more effective than our primal-dual approach?
>
> 3. **Theoretical clarifications.**  Does our theory apply to the hypothesis class of deep neural networks (DNNs), and what is the intuition behind the G-invariance constraint in our formulation?
>
> In our response, we hope to address each of these concerns, as well as several other lingering questions concerning the experiments.
>
> ---
>
> ## Aarbitrary shifts
>
> Your first main comment is that we should more clearly characterize the kinds of shifts that can be handled by our method, i.e.
>
> > "We shouldn't expect to be able to learn transformations between domains for general shifts...The paper could state this more clearly, that the method needs to take advantage of some structure in the transformations."
>
> We agree that there are shifts which do not fall under the umbrella of our approach, and have already updated our draft in line with your comments.  In particular, we have revised Assumption 4.1 to empahsize that our method is designed to handle datasets with a particular, albeit very common, form of dataset shift known as **domain shift** [Storkey, 2009].  This form of shift corresponds to datasets in which the labeling mechanism is invariant to changes under $G$; that is, $P(Y|x) = P(Y|G(x,e))$ for each $x\sim P(x)$ and $\forall e\in\mathcal{E}_\text{all}$.  To this end, we have added a section to our paper that presents a **causal inference** perspective on this assumption.  Our analysis involves characterizing the underlying structural causal model:
>
> $e \rightarrow x^e \leftarrow x \rightarrow y.$
>
> In this model, the domain code $e$ is *independent* of $x$, and $x^e$ is related to $x$ and $e$ via the mapping $x^e = G(x,e)$.  This contextualizes our work in the broader literature, as it clarifies the forms of shifts that our approach is designed to handle.
>
> Notably, while domain shift represents only one of many possible forms of shifts that one might encounter, domain shift is a fundamental issue in appliactions such as medical imaging and autonomous driving.  To quote from a very recent work on this topic [Eastwood et al., 2021]:
>
> > "[Domain] shift is characterized by a change in measurement system, or method of description, and is **particularly pervasive in real-world machine learning systems**. For example, medical imaging systems often fail when deployed to hospitals with different scanners [Zech et al., 2018; AlBadawy et al., 2018; Beede et al., 2020], and self-driving cars often struggle under “shifted” deployment conditions, such as natural variations in lighting [Dai and Van Gool, 2018] or weather conditions [Volk et al., 2019]."
>
> Moreover, an interesting recent work [Ye et al., 2021] shows that many, but not all, standard domain generalization benchmarks exhibit domain shift (see the "diversity shift" axis in Figure 3).  Thus, given that the domain shift assumption is pervasive in the DG literature, methods that can address domain shift are of significant importance to the ML/DG community.  To this end, our empirical evaluations conclusively show that whereas almost all known DG algorithms perform similarly to ERM on domain shift DG datasets, our MBDG algorithm offers significant improvements on several of these datasets.
>
> ---
>
> ## Adversarial training
>
> This is an interesting idea, which also occured to us as we wrote the paper!  Indeed, given the structure induced by Prop. 4.1, our approach does facilitate an AT-based approach. Here are several advantages to using the primal-dual method we propose over AT:
>
> - We can **explicitly characterize the convergence and constraint satisfaction of the primal-dual iteration**, whereas the convergence of AT is only understood in highly simplified settings (see e.g. [Gao et al., 2019]).  More specifically, in Thm. 6.1, we show that if our algorithm runs for sufficiently many steps with small enough step size, it will convergence to a near-minimizer of the primal problem $P^\star$.  We are not aware of how to obtain such a guarantee for AT.
>
> - **Adversarial training is very expensive**, since at each step of an algorithm like PGD, one must compute the gradient with respect to the input of a DNN; in our case, this would involve differentiating through both the classifier $f$ as well as the model $G$.  On the other hand, the primal-dual method requires very little computational overhead.  One must simply compute the violation of the constraints at each iteration, which is significantly less expensive than computing gradients.
>
> - **The primal-dual method allows to to explicitly measure the invariance-based constraint satisfaction** of trained classifiers, whereas it's not clear how to measure this invariance using AT.  Indeed, in Figure 6 in the paper, we measure the $G$-invariance of trained classifiers to get a sense for "how invariant" our classifier is to new domains, which allows one to assess the invariance of a trained classifier.
>
> We believe that a further investigation of AT is definitely an interesting direction for future work, which we plan to pursue in a future manuscript!  However, we feel that this is beyond the scope of this particular paper, as we have already conclusively demonstrated that the primal-dual iteration significantly improves the state-of-the-art in domain generalization for several datasets.
>
> ---
>
> ## Theoretical clarifications
>
> **The VC-dimension of DNNs.** One comment on the theory is that it does not extend to the hypothesis class of DNNs.  We respectfully disagree, because as far as we known, most common classes of DNNs have finite VC-dimension.  Seminal works show that the VC-dimension of DNNs with ReLU non-linearities is $\mathcal{O}(W Llog(W))$, where $W$ is the number of weights and $L$ is the number of layers [Bartlett et al., 2017].  Similar results hold for other non-linearities [Bartlett & Maass, 2002].  Of course, the VC-dimension is is likely quite large for most classes of DNNs, but crucially it is finite, and thus our theory does apply to DNNs.
>
> To address the more general point, our theory can be made to hold **without** the assumption that $\mathcal{H}$ has finite VC-dimension.  The only place we use this assumption is in equation (85) in the proof of Proposition C.1, as it allows us to use off-the-shelf generalization bounds to show that the empirical risk closely approximates the population risk with high probability over the samples.  An alternative approach would be to assume that the so-called **spectral complexity** [Bartlett et al., 2017], which is proportional to the Lipschitz constant, of functions in the hypothesis class is bounded.  As DNNs are known to have bounded Lipschitz constants (see, e.g. [Fazlyab et al., 2019]), the sepctral generalization bounds introduced in [Bartlett et al., 2017] would present an alternative way to prove Theorem 1 without assuming anything about the VC-dimension of $\mathcal{H}$.
>
> **G-invariance.**  Another theoretical question is whether $G$-invariance is stronger than what is needed to solve the minimax problem.  While the $G$-invariance property is strong, we argue that it arises naturally from the domain shift assumption.  More specifically, if a classifier is $G$-invariant, then any change under $G$ should not change the predicted label.  This property can be seen as a deterministic version of the domain shift property $P(y|x) = P(y|G(x,e))$ $\forall e$.  Furthermore, note that this is more or less equivalent to the constraint $f(G(x, e)) = f(G(x, e'))$ $\forall e,e'$ suggested in your review, since the $G$-invariance constraint holds $\forall e$.  Thus, the only difference between $G$-invariance and the constraint suggested in your review is that for $G$-invariance to hold, there must additionally exist at least on $\hat{e}\in\mathcal{E}_\text{all}$ such that $f(x) = f(G(x, \hat{e}))$.  So to summarize, while $G$-invariance is strong, it naturally arises from our (updated) domain shift assumptions and yields strong empirical results.
>
> ---
>
> ## RandAugment + ERM ablation
>
> Regarding the comment:
>
> > "What is the comparison with ERM + RandAugment?"
>
> Thanks for this question.  We want to note that in the documentation for the `Camelyon-WILDS` dataset, the authors of the dataset say that
>
> > "For Camelyon17, models should not use pretraining or color augmentation."
>
> Note that we use neither pretraining nor color augmentation in our approach, since all of our augmentations are **learned directly from data.**  Thus, **using RandAugment with ERM constitutes an unfair comparison with MBDG**, since RandAugment includes augementations such as ColorJitter as well as brightness and contrast adjustments.  Nevertheless, during the rebuttal period, we ran some **new experiments** using ERM + RandAugment, and as the table below shows, while RandAugment improves the performance of ERM significantly, it still lags nearly **9 percentage points behind our approach**.  This demonstrates the utility of learning a domain transformation model from data rather than using fixed augmentations.
>
> | Method | Accuracy |
> |:------:|:--------:|
> | ERM    | 73.3     |
> | ERM  + RandAugment | 86.1 |
> | Ours | 94.8 |

---

> > ### Author Response · Authors · 2021-08-10
> > **Hyperparameter clarification + we will add a conclusion**
> >
> > ## Other lingering questions/comments
> >
> > * Regarding the comment: "A proper comparison would have a sweep over $\lambda$, with some protocol for selecting $\lambda$."  We did sweep over $\lambda$ for MBDG-Reg using twenty randomly selected weights (denoted by $w$ in Algorithm 4) sampled from Unif(0.5, 10) (see Table 4 for details).  The numbers reported for MBDG-Reg in Table 1 reflect the results of this hyperparameter sweep.  Ultimately, a value very close to $\lambda=1$ was selected via hold-one-out cross-validation, which is why for clarity we set $\lambda=1$ for the experiment described in Figure 3.  We want to apologize for this confusion and we intend to clarify this in the final version of the paper.
> >
> > * Regarding the comment: "The range of improvement is pretty wide...and care should be taken when presenting these results faithfully."  Thanks for pointing this out.  We will soften the langauge in our abstract and introduction so as to not overclaim.
> >
> > * Due to spatial constraints, we decided late on in the submission process to cut the conclusion.  As several reviewers have commented on this, if accepted, we fully intend on adding a conclusion to this paper in the camera-ready version (which allows for an additional page in the main body).  We will also add a full description of the PACS dataset and domain generalization task.
> >
> > ---
> >
> > **References**
> >
> > [Storkey, 2009] "When Training and Test Sets Are Different: Characterizing Learning Transfer"
> >
> > [Ye et al., 2021] "OoD-Bench: Benchmarking and Understanding Out-of-Distribution Generalization Datasets and Algorithms"
> >
> > [Gao et al., 2019] "Convergence of adversarial training in overparametrized neural networks"
> >
> > [Sagawa et al., 2019] "Distributionally robust neural networks for group shifts: On the importance of regularization for worst-case generalization"
> >
> > [Robey et al., 2020] "Model-based robust deep learning: Generalizing to natural, out-of-distribution data"
> >
> > [Gulrajani & Lopez-Paz, 2020] "In search of lost domain generalization"
> >
> > [Bartlett & Maass, 2002] "Vapnik-Chervonenkis Dimension of Neural Nets"
> >
> > [Bartlett et al., 2017] "Nearly-tight VC-dimension and pseudodimension bounds for piecewise linear neural networks"
> >
> > [Bartlett et al., 2017] "Spectrally-normalized margin bounds for neural networks"
> >
> > [Fazlyab et al., 2019] "Efficient and accurate estimation of lipschitz constants for deep neural networks"

---

> > > ### Comment · Reviewer_VPwo · 2021-08-14
> > > **response**
> > >
> > > Thanks for the clarifications.
> > > - As one of the reviewers pointed out, the domain shift assumption isn't clearly laid out as the motivation / sketch to get there is a little different as presented in the paper. So this should be reworked. I would note that even within domain shift, there are limitations to what transformations you may be able to learn, as a function of the diversity in the training set domains.
> > > - Thanks for catching the misstatement about VC dimension. Although in general, the DNN generalization bounds like the spectral complexity bound are not very meaningful bounds, anyway - for example they can blow up exponentially in depth.
> > > - The point about G-invariance is good, and might be useful to include in the paper when transitioning to this formulation. For example, one might make the assumption that the environment that the original $x$ comes from is in $\mathcal{E}_{all}$, so that there is some $e$ where $G(x, e) = x$.
> > > - Thanks for running the ERM + RandAugment experiment. From the effectiveness of RandAugment in both ERM and MBDG, it seems like the main shift there is solvable by color jitter, and is somewhat of an easy case for learning the augmentations from data. It's still nice that learning the augmentations helped over the color augmentation baseline.
> > > - I think the main weakness in the results on Camelyon / PACS is that Camelyon seems a little too easy, and that the main improvement on PACS is on sketch. I wouldn't really be sure when to apply MBDG, or just apply some known perturbations to get most of the way there.

---

### Official Review · Reviewer_LPjW · 2021-07-20

**Rating:** 7
**Confidence:** 2

**Summary:**

This paper proposes a new model specifically designed for Out-of-Domain (OOD) detection. The authors start by setting up the domain generalization problem by explaining that (x,y) ~ p_e(x,y) is sampled from different domains ‘e’ in the training and test phase. Making certain assumptions such as (1) the inter-domain variation is only because of p(x), and (2) existing a transformative network g(x,e) that can map observation x to different domains e, the authors propose to slightly modify the problem definition which they call “Model-Based Domain Generalization”. This problem is slightly easier to deal with and under certain conditions, guarantees optimality. The authors finally do an extensive set of experiments to demonstrate the proposed algorithms significantly improves OOD detection results over previous models, as well as the role of regularizing theta with the distance regularizer.

**Limitations And Societal Impact:**

The authors sufficiently discussed the potential impact of OOD.

**Main Review:**

[Overall Comment]

I think this is a good paper. OOD Detection is definitely both an important and popular topic in the field, especially given that, as the authors point out, it’s impact on ML safety. So I certainly think the OOD community can benefit from the findings of this paper. The transition from problem 3.1 to 4.6 certainly makes sense and I think it’s quite elegant. I applaud the authors given that I think this is a lot of content for a single paper, and overall, I think the authors did a good job condensing their theoretical and experimental contributions while ensuring that things are as clear as possible. Given both the methodology and the improvements of the OOD results, I’m happy to recommend acceptance.

[Minor Comments]

-The problem was set-up quite well. I’m not too familiar with the OOD detection field so I found Section 3 quite useful and easy to read (Figure 1 in particular).

-Could the authors comment on the distinction between OOD detection in discriminative models (which is the setting in the current manuscript) and in purely generative models like Flows and VAEs etc?

-While I understand the space constraints, having no conclusion or discussion that summarizes the work is something I would consider a major flaw given the venue, especially given the fact NeurIPS allowed an additional page this year. I will  leave to the AC and other reviews to determine to what extent this is acceptable.

-The supplementary contains the main manuscript as well.

-I think the related work was discussed very well.

[Clarity]

The paper reads well overall. Even though this is a lot of content for a single paper, I think overall the authors did a good job at condensing the materials while making sure that the presented ideas are mostly clear.

Typos:
Page 5, (C3) “We do ‘NOT’ have access to the set of all domains ...”

[Significance]

Given both the the extensive theoretical studies and the improvements of the OOD results, I think the paper can make a valuable contribution to NeurIPS.




**Time Spent Reviewing:**

4

---

> ### Author Response · Authors · 2021-08-10
> **Clarifications regarding generative models + we will add a conclusion**
>
> We're delighted that you liked our paper, and we would like to thank you for your thoughtful review!  We're particularly glad that you thought that we did a reasonable job compressing the theoretical and experimental results and that our problem setup was easy to follow.  We will use this response to tie up the loose-ends and lingering questions in your review.
>
> **OOD detection in generative models.**  We would like to elaborate on this question a bit more to provide context:
>
> > "Could the authors comment on the distinction between OOD detection in discriminative models...and in purely generative models like Flows and VAEs etc?"
>
> To the best of our understanding, OOD detection in (purely) generative models involves assessing the likelihood that generative models assign to OOD samples [Bishop, 1994].  For instance, deep generative models trained on Fasion-MNIST may assign high likelihoods to samples from MNIST, despite the fact that these inputs are clearly OOD [Zhang et al., 2021].  Indeed, it may be an interesting future direction to investigate how such likelihood estimates could be incorporated into our framework.  However, as the reviewer points out, we do not use this generative detection-based method in our approach.  Indeed, our approach is different in the respect that discriminative OOD detection methods use a classifier to estimate a probability distribution over the classes to which a given input might belong [Hendrycks & Gimpel, 2016].  This is in line with our approach, in which generated images are pushed through a classifier, resulting in the OOD accuracies reported in Section 7.
>
> **Lingering questions.**
>
> * Due to space constraints, we decided late on in the submission process to cut the conclusion and parts of the discussion following the experiments.  As several reviewers have commented on this, if accepted, we fully intend on adding a conclusion and further discussion of our results and their implications to this paper in the camera-ready version (which allows for an additional page in the main body).
>
> * Thanks for catching the typo on page 5.  We will correct it.
>
> ---
>
> References:
>
> [Bishop, 1994] " Novelty detection and neural network validation"
>
> [Zhang et al., 2021] "Understanding Failures in Out-of-Distribution Detection with Deep Generative Models"
>
> [Hendrycks & Gimpel, 2016] "A baseline for detecting misclassified and out-of-distribution examples in neural networks"

---

> > ### Comment · Reviewer_LPjW · 2021-08-12
> > **Still happy to recommend acceptance**
> >
> > Thank you for your comments,
> >
> > I have read the authors' response and other reviews.
> >
> > Thank you for the clarification regarding the connection between discriminative and generative models. I do agree that this not really within scope for this paper, but was simply curious to hear the authors' opinion on it.
> >
> > I have read the points raised by reviewer Lmtm as well as the authors' response to it. While I must say that I am not very familiar with this area, the authors' explanation make more sense to me regarding the necessity to make (sometimes strong) assumptions, especially given the impossibility results [Krueger et al., 2021].Regardless of the area, I think it's better to start with assumptions where you can have some gauntness and results, and leave relaxing those assumptions to future work. Therefore, I'm still happy to recommend acceptance.

---

> ### Comment · Reviewer_Lmtm · 2021-08-15
> **post-rebuttal feedback**
>
> I've slightly updated my reviews as well as increasing my the score. All the best!

---

### Official Review · Reviewer_Lmtm · 2021-07-25

**Rating:** 6
**Confidence:** 4

**Summary:**

This paper considers the domain generalization setting where data in all domains can be transformed into one another by a transformation G. Then, authors then propose to learn a predictor invariant to this transformation. Experimental results show some encouraging results.
Overall approach presented is interesting, stimulating and the use of different optimization techniques to formulate the problem is pleasant. However, there is a ‘fatal’ problem in the framework, either by a mathematical mistake (i.e., moving from Eq 31 to 32 in the appendix), or authors have failed to declare a very strong assumption that that p(y|x) = p(y | G(x,e)) (see detailed comments below). If this point can be clarified, I would be willing to reconsider my rating.

----- post-rebuttal
I want to thank the reviewer for their thorough rebuttal. While I still want to see some solid analysis on the effect of the transform G and the assumption, the new discussions made sense and early empirical results on testing far domains are encouraging. The authors, please, make sure you factor them into the paper as they will not only enhance the understanding but also increase the validity of your work.


**Limitations And Societal Impact:**

The author didn’t adequately address the limitation of the transformation G. Moreover, since there is an implicit assumption, this method might not work properly in some situation (shifted label marginals)

**Main Review:**

The authors implicitly assume that: after transport x ~ P(x) by transformation G( . , e), the labeling mechanism is unchanged, i.e., P(Y | x) = P(Y | G(x,e)) (Proposition 4.3)

This assumption allows authors to sample from a common distribution, i.e., (x, y) ~ P(x, y) instead of sample directly from distribution of each domain, i.e., (x, y) ~ P^e(x, y) (Proposition 4.3, equation 32). Concretely, we will derive an equivalent sampling process to (x, y) ~ P^e(x, y) as follow:
(x, y) ~ P^e(x, y) can be cast into: first x ~ P^e(x), then y ~ P^e(Y|x) (Proposition 4.3, equation 28)
Due to Assumption 4.1, P^e(Y|x) = P(Y|x) for all x, the sampling process is: first x ~ P^e(x), then y ~ P(Y|x) (Proposition 4.3, equation 30)
Due to Assumption 4.2, there is some G(. , e) that transports P(x) to P^e(x). This means the sampling process x ~ P^e(x) can be replaced by x ~ P(x), then passing G(x, e) to the label generative process. Therefore, the labeling mechanism must be y ~ P(Y| G(x, e)). The whole process is x ~ P(x), y ~ P(Y| G(x, e)), and is equivalent to (x, y) ~ P^e(x, y).
However, the authors instead sample label as y ~ P(Y| x) with x is from P(x), and G(x, e) is only used in the loss function (Proposition 4.3, equation 30, inner-loop expectation). This leads to sampling process x ~ P(x), y ~ P(Y| x) is not equivalent to x ~ P(x), y ~ P(Y| G(x, e)) due to the change in labeling mechanism.

Authors’ derivation can work under further assumption that P(Y|X) is G-invariant: “P(Y| x) = P(Y| G(x, e)) for all x and e”. However, this is a rather strong assumption, for example it doesn’t hold when there is a large shift in label marginals. Note that this requirement is an additional assumption, and is not coincide with Assumption 4.1, since Assumption 4.1 says all domains e must use the same set of condition distributions, i.e., P^e(Y|x) = P(Y|x) for all x, but it doesn’t require P(Y|x) = P(Y|x’) for 2 different x, x’.

To illustrate further, let R be the data space and {-1, 1} be the label set. Moreover, all domains P^e(x,y) must share the same conditional P(Y|X), that if X<0, assign Y = -1, and if X>0, assign Y=1. Considering P^e(x, y) with P^e(x) = Uniform[0,1], we have for all x ~ P^e(x), label y=1. Furthermore, let P(x,y) be another distribution with P(x) = Uniform[-1,0], then for all x ~ P(x), label y = -1. Next, we will find a sampling process equivalent to (x, y) ~ P^e(x, y). We can first sample from x ~ P(x), and then shift data by G(x) = x+1 to obtain P^e(x). However, if we label by y ~ P(Y|x), all labels will be -1, because X<0 and P(Y = -1 | X<0) =1, and hence we do not obtain the equivalence of (x, y) ~ P^e(x, y). The correct sampling process must be x ~ P(x), transport to G(x), and then label y ~ P(Y|G(x)), which will now return y=1, since G(x)>0 and P(Y = 1 | X>0) =1. This toy experiment shows that this paper’s approach might fail in situations where domains have different label margins.

Using rather unrealistic assumption that labeling mechanism is unchanged after transforming x into G(x, e), for all x ~ P(x) and for all domain e, authors define the key optimization problem (Problem 4.6) in Proposition 4.5, which seek to find a function f that is also invariant to transformation G, i.e., f(x) = f(G(x, e)). However, as demonstrated before, assumption “P(Y| x) = P(Y| G(x, e)) for all x and e” is unrealistic, therefore this optimization problem doesn’t guarantee to reflect real datasets.

Generalization of G is critical to generalization of f, which is expressed in the infinite-dimensional constraint term in the Lagragian. However, in the empirical version, this constraint is approximated by summing over training domains, which is usually of small number in most real datasets. Therefore, G could learn to transform between these training domains, but might not be able to do so for target domains, notably target domains of large dissimilarity to training domains, because G is a map between high-dimensional input space, trained using only a small number of domains. Therefore, for these “distant” target domains, it is unlikely that f generalizes, and f can only work on target domains closely similar to training domains. It is therefore interesting to see (1) the performance of predictor f in the case where the target domain is very different to each training domain, (2) performance of this method on other datasets with more diverse domains, and (3) how the number of training domains affects the generalization of f.

The form of distance d is not discussed.
Typos and other errors
Line 217, P(\mathcal{E}_{all}) is not explicitly defined.
Line 926, expectation over P^e(x) is missing
Line 973, eq 54, drop expectation over risk difference




**Time Spent Reviewing:**

12

---

> ### Author Response · Authors · 2021-08-10
> **Clarifying our assumptions on the labelling mechanism + updates to our paper to incorporate your feedback**
>
> **Key points in our response:**   Thank you for writing this meticulous review of our work!  We're glad that you find our approach "interesting" and "stimulating" and that you feel that "the use of different optimization techniques to formulate the problem is pleasant."  Based on your review, it seems that you have one major concern and several other minor concerns/loose ends.
>
> * **Major concern.**  Your major concern has to do with whether there is an implicit assumption -- which is known in the literature as **domain shift** -- involving the label mechanism $P(Y|x)$, and whether this assumption is realisitic for real-world datasets.  This is nicely illustrated with a simple binary classification example.
>
> * **Minor concerns.**  Another concerns is regarding whether the predictor $f$ can generalize to domains that are "far" from the training domains, and whether we can expect $G$ to capture these "far" domains.  There are also some loose ends, including a question concerning the precise form of the metric $d$.
>
> In our detailed comments below, we address each of these concerns.  In particular, with respect to your main concern, we want to emphasize that we agree that this assumption is needed and should be made explicit, and that we have already updated our paper to reflect your feedback.  To this end, in the detailed comments, we argue that while this so-called domain shift assumption is relatively strong,
>
> 1. restrictions and/or assumptions are necessary in the domain generalization (DG) setting, as generalizing to arbitrarily-shifted test domains in $\mathcal{E}_\text{all}$ is impossible,
> 2. numerous standard ML/DG benchmarks exhibit domain shift, and moreover domain shift in these datasets is known to significantly degrade the performance of state-of-the-art classifiers, and
> 3. as far as we know, our algorithm is the first to improve OOD generalization under domain shift in the setting of DG.
>
> Please let us know whether our response clears up your questions, and if so, since we’ve done our best to address your concerns and updated our paper, we hope that you would consider increasing your score.
>
> ---
>
> ## Major concerns -- detailed comments
>
> **Assumption on P(Y|x).**  We agree with the reviewer that our derivations implicitly use the condition
>
> $P(Y|x) = P(Y|G(x,e)) \quad\text{a.e. } x\sim P(x) \quad\forall e\in\mathcal{E}_\text{all},$
>
> an assumption which in past work has been referred to as **domain shift** [Storkey, 2009, Heinze-Deml & Meinshausen, 2017, Ahmed et al., 2021] (and less commonly as "label-invariant transformations" in [Wu et al., 2020] and "general perturbations" in [Wong & Kolter, 2020]).  In essence, the domain shift assumption means that transforming instances $x$ under $G$ should not change the label $y$ of any instance.  It was not our intention to suppress this assumption, and shortly after submitting this paper to NeurIPS, we updated our draft -- in particular Assumption 4.1 -- to state and explain this domain shift assumption.  In particular, we have added a section that presents a **causal inference** perspective on this assumption.  Our analysis involves characterizing the underlying structural causal model, which looks like this
>
> $e \rightarrow x^e \leftarrow x \rightarrow y.$
>
> In this model, the environment code $e$ is *independent* of $x$, and $x^e$ is related to $x$ and $e$ via the domain transformation mapping $x^e = G(x,e)$.  This contextualizes our work in the broader literature, as it clarifies the forms of shifts that our approach is designed to handle.Moreover, we have carefully checked the proofs, and the reviewer is correct that all of the theorems still hold under the domain shift assumption.  A final version of the paper will reflect these changes, and we thank the reviewer for identifying this source of confusion.
>
> **Further assumptions are necessary for DG.**  One of your primary concerns is whether the domain shift assumption is too strong.  To this end, we want to emphasize a commonly-accepted principle in the DG literature: in general, generalization to arbitrarily-shifted test distributions in $\mathcal{E}_\text{all}$ is impossible [Krueger et al., 2021].  Therefore, we argue that it is **necessary** to place assumptions on the data-generating mechanism if one is to prove anything regarding the ability of a predicter to generalize OOD.  We believe that without designing methods that are suitable for particular shifts, it may be impossible to design algorithms that consistently beat ERM, as evinced in a recent large-scale empirical study [Gulrajani & Lopez-Paz, 2020].
>
> **The domain shift assumption is pervasive in applications.**  Although the domain shift assumption is relatively strong and does not apply to every possible DG setting, a wide variety of datasets and settings do satisfy the domain shift assumption, and moreover it has recently been argued that domain shift is a fundamental problem in the OOD setting.  To quote from [Eastwood et al., 2021]:
>
> > "[Domain] shift is characterized by a change in measurement system, or method of description, and is **particularly pervasive in real-world machine learning systems**. For example, medical imaging systems often fail when deployed to hospitals with different scanners [Zech et al., 2018; AlBadawy et al., 2018; Beede et al., 2020], and self-driving cars often struggle under “shifted” deployment conditions, such as natural variations in lighting [Dai and Van Gool, 2018] or weather conditions [Volk et al., 2019]."
>
> Indeed, the domain shift assumption underlies a wide range of problems, including:
>
> * **Invariance to rotations and other geometric transformations**.  For numerous datasets (e.g. `RotatedMNIST` [Ghifary et al., 2015]), rotating images or transforming them by other group-based transformations (e.g. scaling or shifting an image) is assumed to have no impact on the labelling mechanism [Engstrom et al., 2019].
>
> * **Medical imaging applications**. Imaging equipment naturally introduces domain shift.  In this setting, $x\sim P(x)$ could be a group of cells which is being assessed for cancerous tumors, and $x^e = G(x,e)$ can be thought the observation of those cells using the equipment available at a particular hospital $e\in\mathcal{E}_\text{all}$ (see the `Camelyon17-WILDS` dataset [Koh & Sagawa et al., 2021]).  The underlying assumption here is that the imaging device should **not** change whether or not cancerous cells are present, which is equivalent to domain shift.
>
> * **Autonomous vehicles**.  A typical application in autonomous vehicles is to classify street signs (see the `CURE-TSR` dataset in [Temel et al., 2017]).  Oftentimes, distributional shifts arise in such tasks due to shifting weather conditions such as snow or rain, resulting in significant performance degradation [Robey et al., 2020].  Such shifts fall under the domain shift assumption, as weather conditions do not change the label of the street signs.  Recently, the `Shifts OOD dataset` was proposed precisely for this problem in the 2021 NeurIPS competition track [Malinin et al., 2021].
>
> * **Robustness to norm-bounded perturbations**.  In this case, domain shift is due to a *linear* model $G(x,e) = x + e$, and one assumes that $P(Y|x) = P(Y|x+e)$ $\forall e$ in a norm ball cenetered at the origin [Biggio et. al, 2013, Szegedy et al., 2013, Madry et al., 2017].
>
> As shown in [Ye et al., 2021], many standard DG benchmarks, including `TerraIncognita` [Beery et al., 2018], `Camelyon17-WILDS` [Koh & Sagawa et al., 2021], `PACS` [Yang et al., 2017], `ImageNet-Sketch` [Wang et al., 2019], `ColoredMNIST` [Arjovsky et al., 2019], and `OfficeHome` [Venkateswara et al., 2017], are all characterized by domain shift (see the "diversity shift" axis in Figure 3 of [Ye et al., 2021]).
>
> **Empirical improvements for domain shift.**  Given that the domain shift assumption is pervasive both within and outside of the field of DG, methods that can address domain shift are of significant importance to the ML community.  To this end, our empirical evaluations conclusively show that whereas almost all known DG algorithms perform similarly to ERM on domain shift DG datasets, our MBDG algorithm offers significant improvements on these datasets.
>
> [our response is continued in the next comment]

---

> > ### Author Response · Authors · 2021-08-10
> > **New experiments concerning "far" domains + clarifications on the form of the distance metric**
> >
> > ## Minor concerns -- detailed comments
> >
> > **Generalizing to "far" domains.**  In our framework, it is true that $f$ can only be expected to generalize to domains that are generated by $G$.  For some datasets, e.g. `Camelyon17-WILDS`, it is possible to learn the extent of the expected distributional shifts from relatively few, similar domains, but in general, more diverse domains and/or a large number of domains may be required.  We think that each of the suggested diretions in this part of your review are critical directions for future work, and we hope to address these issues in a future manuscript.
> >
> > As a first step toward addressing these concerns, during the rebuttal period we designed an experiment in which the training domains are "farther" apart from the test domain.  In particular, we introduce a new DG setting that uses data from `ImageNet` and `ImageNet-c`.  `ImageNet-c`, which was recently curated in [Hendrycks & Diettrich, 2019].  `ImageNet-c` contains numerous copies of `ImageNet`, each of which is transformed by a different naturally-occurring corruption; these corruptions include shifts in brightness, contrast, and various weather conditions.  Given these datasets, we consider a setting in which the training domains consist: a labeled dataset from classes 10-59 of `ImageNet`, and two domains containing data from classes 0-9 drawn from two different subsets of `ImageNet-c`.  In this task, the test domain consists of labeled data from classes 10-59 of `ImageNet` that has been subjected to both of the corruptions used in the `ImageNet-c` training domains.  For example, in row 1 of the table shown below, we consider a setting in which the training domains contain labeled `ImageNet` data, and two domains from `ImageNet-c`, where one subset is corrupted by a shift in contrast and the other is corrupted by a shift in fog.  Thus, in this example, the test domain would contain images corrupted by **simultaneous shifts** in contrast and fog.
> >
> > Note that this setting captures "farther" shifts in two ways: firstly, the two shifts that characterize the test domain are not simultaneously present in the training domains (i.e. the classifier nevers sees data with both contrast and fog shifts); secondly, the data corresponding to the shifts in `ImageNet-c` belong to different classes than the data in the test domain.  Altogether, this engenders a highly challenging DG setting.  In the table below, we report the classification accuracy for MBDG, ERM, and IR.  For all of the combinations of challenges we considered, MBDG significantly outperformed the baselines.   While this is only a first step toward answering these questions regarding "far" domains, we plan to build on these results significantly in future work.
> >
> > | ImageNet-c Corruptions | ERM | IRM | MBDG |
> > | :--------------------: | :-: | :-: | :-: |
> > | Contrast & Fog | 8.40 | 8.16 | **23.2** |
> > | Contrast & Brightness | 13.6 | 11.7 | **49.9** |
> > | Brightness & Snow | 53.3 | 48.9 | **58.3** |
> > | Brightness & Fog | 50.3 | 45.7 | **58.8** |
> >
> > ---
> >
> > **Other loose ends**
> >
> > * **The form of $d$.**  The form of $d$ used in practice was somewhat buried in the appendix -- see line 1072 in Appendix E.  Specifically, for the experiments, we using the KL-divergence for $d$, although other choices are possible.  We will move this description to the main text.
> >
> > * **Typos.** Thanks for catching these.  We will fix them in the final version.
> >
> > ---
> >
> > **References**
> >
> > [Krueger et al., 2021] Out-of-distribution generalization via risk extrapolation (rex)
> >
> > [Storkey, 2009] "When Training and Test Sets Are Different: Characterizing Learning Transfer"
> >
> > [Gulrajani & Lopez-Paz, 2020] "In search of lost domain generalization"
> >
> > [Heinze-Deml & Meinshausen, 2017] "Conditional variance penalties and domain shift robustness"
> >
> > [Ahmed et al., 2021] "Systematic generalisation with group invariant predictions"
> >
> > [Wu et al., 2020] "On the generalization effects of linear transformations in data augmentation"
> >
> > [Wong & Kolter, 2020] "Learning perturbation sets for robust machine learning"
> >
> > [Eastwood et al., 2021] "Source-Free Adaptation to Measurement Shift via Bottom-Up Feature Restoration"
> >
> > [Zech et al., 2018] "Variable generalization performance of a deep learning model to detect pneumonia in chest radiographs: a cross-sectional study"
> >
> > [AlBadawy et al., 2018] "Deep learning for segmentation of brain tumors: Impact of cross‐institutional training and testing"
> >
> > [Beede et al., 2020] "A human-centered evaluation of a deep learning system deployed in clinics for the
> > detection of diabetic retinopathy"
> >
> > [Dai and Van Gool, 2018] "Dark model adaptation: Semantic image segmentation from daytime to nighttime"
> >
> > [Volk et al., 2019] "Towards robust CNNbased object detection through augmentation with synthetic rain variations"
> >
> > [Biggio et. al, 2013] "Evasion attacks against machine learning at test time"
> >
> > [Szegedy et al., 2013] "Intriguing properties of neural networks"
> >
> > [Madry et al., 2017] "Towards deep learning models resistant to adversarial attacks"
> >
> > [Ghifary et al., 2015] "Domain generalization for object recognition with multi-task autoencoders"
> >
> > [Engstrom et al., 2019] "Exploring the Landscape of Spatial Robustness"
> >
> > [Koh & Sagawa et al., 2021] "Wilds: A benchmark of in-the-wild distribution shifts"
> >
> > [Temel et al., 2017] "CURE-TSR: Challenging unreal and real environments for traffic sign recognition"
> >
> > [Robey et al., 2020] "Model-based robust deep learning: Generalizing to natural out-of-distribution data"
> >
> > [Malinin et al., 2021] "Shifts: A Dataset of Real Distributional Shift Across Multiple Large-Scale Tasks"
> >
> > [Ye et al., 2021] "OoD-Bench: Benchmarking and Understanding Out-of-Distribution Generalization Datasets and Algorithms"
> >
> > [Beery et al., 2018] "Recognition in terra incognita."
> >
> > [Yang et al., 2017] "Deeper, broader and artier domain generalization"
> >
> > [Wang et al., 2019] "Learning Robust Global Representations by Penalizing Local Predictive Power"
> >
> > [Arjovsky et al., 2019] "Invariant risk minimization"
> >
> > [Venkateswara et al., 2017] "Deep hashing
> > network for unsupervised domain adaptation"
> >
> > [Hendrycks & Diettrich, 2019] "Benchmarking Neural Network Robustness to Common Corruptions and Perturbations"

---

### Decision · Program_Chairs · 2021-09-27

**Decision:**

Accept (Poster)

**Comment:**

The reviewers reached a consensus that the paper has an interesting and novel idea and achieved good improvements over prior works on many (though not all) benchmark datasets. The AC recommends acceptance.